# Neuro-evolutionary evidence for a universal fractal primate brain shape

**Yujiang Wang[1,2,3]\*, Karoline Leiberg[1], Nathan Kindred[2], Christopher R Madan[4], Colline Poirier[2], Christopher I Petkov[2,5], Peter Neal Taylor[1,2,3], Bruno Mota[6]**

[1]CNNP Lab (https://www.cnnp-lab.com), School of Computing, Newcastle University, Newcastle upon Tyne, United Kingdom; [2]Faculty of Medical Sciences, Newcastle University, Newcastle upon Tyne, United Kingdom; [3]UCL Institute of Neurology, Queen Square, London, United Kingdom; [4]School of Psychology, University of Nottingham, Nottingham, United Kingdom; [5]Department of Neurosurgery, University of Iowa, Des Moines, United States; [6]metaBIO Lab, Instituto de Física, Universidade Federal do Rio de Janeiro (UFRJ), Rio de Janeiro, Brazil

**\*For correspondence:**
yujiang.wang@newcastle.ac.uk

**Competing interest:** The authors declare that no competing interests exist.

**Abstract** The cerebral cortex displays a bewildering diversity of shapes and sizes across and within species. Despite this diversity, we present a universal multi-scale description of primate cortices. We show that all cortical shapes can be described as a set of nested folds of different sizes. As neighbouring folds are gradually merged, the cortices of 11 primate species follow a common scale-free morphometric trajectory, that also overlaps with over 70 other mammalian species. Our results indicate that all cerebral cortices are approximations of the *same* archetypal fractal shape with a fractal dimension of $d_f = 2.5$. Importantly, this new understanding enables a more precise quantification of brain morphology as a function of scale. To demonstrate the importance of this new understanding, we show a scale-dependent effect of ageing on brain morphology. We observe a more than fourfold increase in effect size (from two standard deviations to eight standard deviations) at a spatial scale of approximately 2 mm compared to standard morphological analyses. Our new understanding may, therefore, generate superior biomarkers for a range of conditions in the future.

## eLife assessment

This study presents **valuable** framework and findings to our understanding of the brain cortex as a fractal object. Based on detailed methodology, the evidence provided on the stability of its shape property within 11 primate species is **convincing**, as well as the scale-specific effects of ageing on the human brain. This study will be of interest to neuroscientists interested in brain morphology, and to physicists and mathematicians interested in modeling the shapes of complex objects.

## Introduction

The morphological complexity of the mammalian cerebral cortex has fascinated scientists for generations, with cortices across and within species, exhibiting a large diversity of shapes and sizes. Such diversity is not arbitrary, however. The mammalian brain folds into stereotypical, hierarchically-organised structures such as lobes and major gyri. In fact, qualitative and quantitative regularities in cortical scaling have often been suggested and observed (*Zhang and Sejnowski, 2000*; *Francis et al., 2009*; *Karbowski, 2011*; *Mota and Herculano-Houzel, 2014*). More specifically, through modelling the mechanism of cortical folding from a statistical physics approach, we have previously derived a theoretical scaling law relating pial surface area $A_t$, exposed surface area $A_e$ (the exposed surface area can be thought of as the surface area of a piece of cling film wrapped around the brain;

**eLife digest** Many of the brain's essential functions – from decision-making to movement – take place in its outer layer known as the cerebral cortex. The shape of the cerebral cortex varies significantly between species. For instance, in humans, it is folded in to grooves and ridges, whereas in other animals, including mice, it is completely smooth. The structure of the cortex can also differ within a species, and be altered by aging and certain diseases.

This vast variation can make it difficult it to characterize and compare the structure of the cortex between different species, ages and diseases. To address this, Wang et al. developed a new mathematical model for describing the shape of the cortex.

The model uses a method known as coarse graining to erase, or 'melt away', any cortical folds or structures smaller than a given threshold size. As this threshold increases, the cortex becomes progressively smoother. The relationship between surface areas and threshold sizes indicates the fractal dimension – that is, how fragmented the cortex is across different scales.

Wang et al. applied their model to the brain scans of eleven primates, including humans, and found the fractal dimension of the cortex was almost exactly 2.5 for all eleven species**.** This suggests that the cortices of the different primates follow a single fractal shape, which means the folds of each cortex have a similar branching pattern. Although there were distinctions between the species, they were mainly due to the different ranges of fold sizes in each cortex. The model revealed that the broader the range of fold sizes, the more folded the brain – but the fractal pattern remains the same.

The brain melting method created by Wang et al. provides a new way to characterise cortical shape. Besides revealing a hitherto hidden regularity of nature, they hope that in the future their new method will be useful in assessing brain changes during human development and ageing, and in diseases like Alzheimer's and epilepsy.

mathematically, for the remaining paper it is the convex hull of the brain surface), and average cortical thickness $T$:

$$A_t T^{\frac{1}{2}} = k A_e^{\frac{5}{4}}.$$

(1)

This scaling law, relating powers of cortical thickness and surface area metrics, was shown to be valid across mammalian species (*Mota and Herculano-Houzel, 2015*) and within the human species (*Wang et al., 2016*), as well as to the structures and substructures of individual brains (*Wang et al., 2019*; *Leiberg et al., 2021*). Notably, across all these cases the dimensionless offset $k$ is shown to be near invariant. However, this universality, presumed to be arising from universal physical principles and evolutionarily conserved biomechanics, says little about what that cortical shape actually is, beyond a constraint binding three morphometric parameters. In this paper, we take *Equation 1* as an empirical starting point to create a new and hierarchical way of expressing cortical shape. Specifically, we introduce a coarse-graining procedure that renders the cortex at different spatial scales, or resolutions. We show that *coarse-grained primate cortices at each spatial scale can be understood as approximations of the same universal self-similar archetypal form*, of which the observed scaling law *Equation 1* can then be shown to be a direct consequence.

Besides revealing a symmetry in nature hidden under much apparent complexity, our results indicate a conservation of morphological relationships across evolution. We will show that these results further provide us with a new and powerful tool to express and analyse cortical morphology. As an example, we will calculate the effects of human ageing across spatial scales and show that the effects are highly scale-dependent.

## Mathematical background

The universal scaling law *Equation 1* can be rewritten in a suggestive way

$$\frac{A_t}{A_0} = \left(\frac{A_e}{A_0}\right)^{1.25},$$

(2)

where the $A_0 = \frac{T^2}{k^4}$ is a fundamental area element that defines the threshold between gyrencephaly (folded cortex) and lissencephaly (smooth cortex) when $A_t = A_e = A_0$. For a constant $k$ the value of $A_0$ is a multiple of $T^2$, indicating that cortical thickness determines the size of the smallest possible gyri and sulci.

This re-writing highlights a new perspective, or interpretation of the scaling law: it now suggests a relationship between intrinsic and extrinsic measures of cortical size (given the folded laminar structure of the cortex, areas are the more natural way of measuring its 'size'), $A_t$ and $A_e$, respectively, measured in units of $A_0$. This is reminiscent of fractal scaling (**Mandelbrot, 1983**), where a complex shape reveals ever smaller levels of self-similar detail as it is probed in ever smaller scales (or equivalently, higher resolutions), represented here by $A_0$. The scaling, or power exponent between the measured intrinsic and extrinsic sizes is the so-called fractal dimension.

Although actual fractals are mathematical abstractions, they can often be defined as the limit of iterative processes. Many structures in nature, and in particular biology (**Elston and Zietsch, 2005**; **Codling et al., 2008**; **Ionescu et al., 2009**; **Losa, 2011**; **Klonowski, 2016**; **Di Ieva, 2016**; **Reznikov et al., 2018**), are good approximations of a fractal. *Equation 2* is suggestive, but not proof, that cortices are among these forms, with a fractal dimension of $1.25 \times 2 = 2.5$ (the factor 2 being the topological dimension of areas). Indeed, fractal scaling for various aspects of cortical morphology has often been postulated (**Free et al., 1996**; **Kiselev et al., 2003**), with a number of recent papers making use of MRI data (**Marzi et al., 2021**; **Jao et al., 2021**; **Meregalli et al., 2022**; **Díaz Beltrán et al., 2024**). Most recently published estimates of fractal dimension for the whole cortex are indeed close to 2.5 (**King et al., 2010**; **Madan and Kensinger, 2016**; **Madan and Kensinger, 2017**; **Marzi et al., 2020**).

Here, for the first time, we propose to directly construct morphologically plausible realisations of cortices at any specified spatial scale, or resolution. This is achieved through a coarse-graining method that removes morphological details smaller than a specified scale while preserving surface integrity. For example, at a set scale of 3 mm, sulcal walls that are less than 3 mm apart would be removed, and the neighbouring gyri would be fused. This method is a new systematic way of obtaining shape properties from the cortex in terms of a sequence of morphometric measurements as spatial scale varies. By examining how areas scale across coarse-grained versions of actual primate cortices, we will be able to directly verify cortical self-similarity.

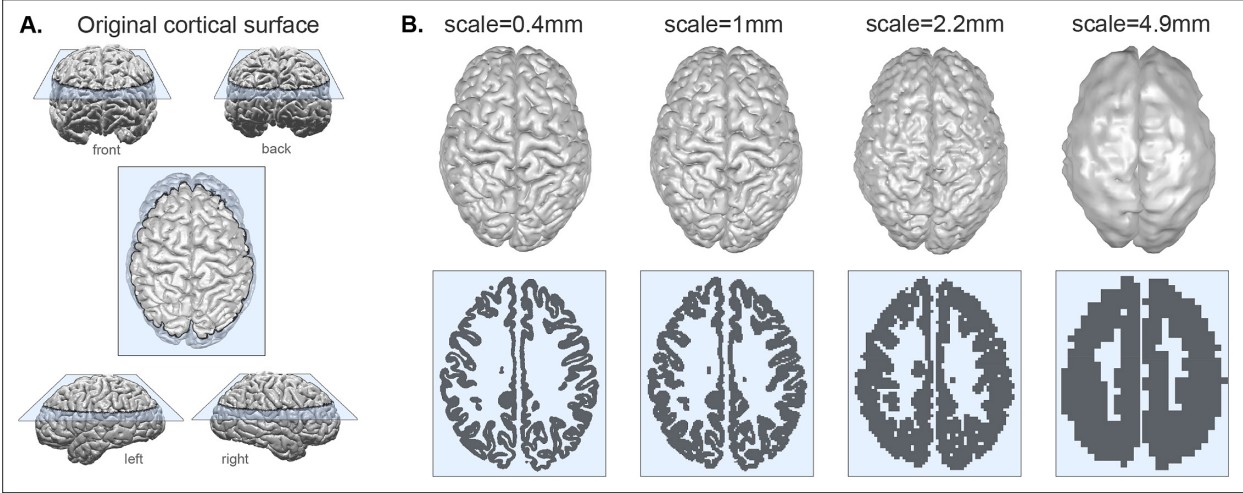

**Figure 1.** Coarse-graining a cortex at different scales. (**A**) Example of original pial surface from a healthy human viewed from varying angles. (**B**) Same example cortical surface from A, coarse-grained to different spatial scales. Top row shows the resulting pial surfaces (with a small amount of smoothing applied for visualisation purposes here). Bottom row shows the corresponding voxelisation at each scale through the slice indicated by the blue plane in panel A. Note that the actual size of the brains for analysis is rescaled (see Methods and *Figure 2*); we display all brains scaled at an equal size here for the ease of visualisation of the method.

## Method

### Coarse-graining method

As a starting point for a coarse-graining method, we suggest to turn to a well-established method that measures a fractal dimension of objects: the so-called box-counting algorithm (). Briefly, this algorithm fills the object of interest (the cortex in our case) with boxes, or voxels of increasingly larger sizes and counts the number of boxes in the object as a function of box size. As the box size increases, the number of boxes decreases; and in a log-log plot, the slope of this relationship indicates the fractal dimension of the object. In our case, this method would not only provide us with the fractal dimension of the cortex, but, with increasing box size, the filled cortex would also contain less and less detail of the folded cortex. Intuitively, with increasing box size, the smaller details below the resolution of a single box would disappear first, and increasingly larger details will follow – precisely what we require from a coarse-graining method. We, therefore, propose to expand the traditional box-counting method beyond its use to measure the fractal dimension, but to analyse the reconstructed cortices as different realisations of the original cortex at the specified spatial scale.

Concretely, our proposed method requires the bounding pial and white matter surfaces of the cortical ribbon as input. We obtained these surfaces based on reconstructions from magnetic resonance imaging data in 11 different primate species. Algorithmically, we then segment the space between the original pial and white matter surfaces into a 3D grid of boxes of the desired scale $\lambda$, where each box is a cube of dimensions $\lambda \times \lambda \times \lambda$. We also term the 3D grid of cubes 'voxelisation,' as it effectively captures the cerebral cortex as voxels in 3D space (*Figure 1B* bottom row). At any given scale, or voxel size, this process effectively erases morphological features (folds) that are smaller than the cube size. Visually, increasing the voxel size appear as if the cortex is 'melting' and 'thickening' (*Figure 1*, and videos: https://bit.ly/3CDoqZQ).

A more technical and detailed description and discussion of the algorithm is provided in Appendix 1. Note this method has also no direct dependency on the original MR image resolution, as the inputs are *smooth* grey and white matter surface meshes reconstructed from the images using strong (bio-) physical assumptions and, therefore, containing more fine-grained spatial information than the raw images (see also Appendix 2).

### Rescaling coarse-grained outputs for analysis

Morphological properties, such as cortical thicknesses measured in our 'melted' brains are to be understood as a thickness relative to the size of the brain. Therefore, to analyse the scaling behaviour of the different coarse-grained realisations of the same brain, we apply an isometric rescaling process that leaves all dimensionless shape properties unaffected (more details in Appendix 3.1). Conceptually, this process fixes the voxel size, and instead resizes the surfaces relative to the voxel size, which ensures that we can compare the coarse-grained realisations to the original cortices, and test if the former, like the latter, also scales according to *Equation 1*. Resizing, or more precisely, shrinking the cortical surface is mathematically equivalent to increasing the box size in our coarse-graining method. Both achieved an erasure of folding details below a certain threshold. After rescaling, as an example, the cortical thickness also shrinks with increasing levels of coarse-graining, and never exceeds the thickness measured at native scale.

### Independent morphological measures of shape

To better characterise the coarse-grained cortices in terms of their similarity in offset, we use a previously introduced (*Wang et al., 2021*) a set of independent measures, $K$, $I$, and $S$, that summarise the morphometry of the cortex in a natural and statistically robust way. In this framework, isometrically scaled copies of the same morphometry all map onto a line along the $I = \log A_t + \log A_e + \log T^2$ direction, which is perpendicular to a $K \times S$ plane that fully summarises their shape. $K = \log A_t - \frac{5}{4} \log A_e + \frac{1}{4} \log T^2$ is the direction defined by the offset $k$ of the scaling law *Equation 1*, while direction $S = \frac{3}{2} \log A_t + \frac{3}{4} \log A_e - \frac{9}{4} \log T^2$ captures the remaining information about shape, and can be regarded as a simple measure of morphological complexity. *Wang et al., 2021* provides a detailed derivation and demonstration of the superior sensitivity and specificity of these new morphometric measures. The advantage of using this framework here is that we can assess the offset $K$ (and shape term $S$) without interference by isometric size effects, including any re-scaling procedures.

## Data and processing

With the exception of the marmoset data, all other cortical surface reconstructions were based on healthy individual brains.

### Human data

To study healthy human adults, we used the Human Connectome Project (HCP) MRI data, available at https://db.humanconnectome.org/; *Van Essen et al., 2012*, obtained using a 3T Siemens Skyra scanner with 0.7 mm isotropic voxel size. We used the HCP minimally pre-processed FreeSurfer data output, which provided the pial and white matter surface meshes we required. We selected five random subjects in the age category 22–25 y.o. and show one example subject (103414) in the main text, and the remaining subjects in Appendix 3.

To study the alterations associated with human ageing, we used T1 and T2 weighted MRI brain scans from The Cambridge Centre for Ageing and Neuroscience (Cam-CAN) dataset (available at http://www.mrc-cbu.cam.ac.uk/datasets/camcan/ *Shafto et al., 2014*; *Taylor et al., 2017*). Cam-CAN used a 3T Siemens TIM Trio System with 1 mm isotropic voxel size (for more details see *Shafto et al., 2014*; *Taylor et al., 2017*). From the Cam-CAN dataset we retained 644 subjects that successfully completed preprocessing (with Freesurfer recon-all) without errors. From these subjects, we selected all subjects between the ages of 17–25 inclusive (forming the 20 y.o. cohort, n=27); we also selected all subjects between the ages of 77–85 inclusive (forming the 80 y.o. cohort, n=86).

To confirm the ageing results, we also obtained an independent dataset from the Nathan Kline Institute (NKI)/Rockland sample (*Nooner et al., 2012*; http://fcon_1000.projects.nitrc.org/indi/pro/nki.html) using the same procedure as described for the CamCAN dataset.

The MR images of both CamCAN and NKI datasets were first preprocessed by the FreeSurfer 6.0 pipeline *recon-all*, which extracts the grey-white matter boundary as well as the pial surface. These boundaries were then quality checked by visual inspection for particularly the young and old cohorts and manually corrected where needed.

For all three datasets, we obtained the pial and white matter surfaces for further analysis. In the current work, the analysis is always hemisphere-based, as in our previous work (*Mota and Herculano-Houzel, 2015*; *Wang et al., 2016*). We did not perform a more regionalised analysis, which is also possible (*Wang et al., 2019*; *Leiberg et al., 2021*). Future work using the principle demonstrated here can be directly extended to derive regionalised measures across scales.

### Non-human primate data

#### Macaque

Rhesus Macaque MRI scans were carried out at the Newcastle University Comparative Biology Centre. Macaques were trained to be scanned while awake and sat in a primate chair. Both T1-weighted MP-RAGE and T2-weighted RARE sequences were acquired, using a vertical MRI scanner (Biospec 4.7 Tesla, Bruker Biospin, Ettlingen, Germany).

Scans were processed using a custom macaque MRI pipeline, incorporating ANTs, SPM, FreeSurfer, and FSL. Briefly, this involved the creation of precursor mask in SPM, denoising (ANTs DenoiseImage) and debiasing (ANTs N4BiasFieldCorrection, and Human Connectome BiasFieldCorrection script), creation of a final mask in SPM and then processing in FreeSurfer (using a modified version of the standard FreeSurfer processing pipeline). Reconstructed pial and white matter surfaces were visually quality-controlled in conjunction with the MR images.

#### Marmoset

The marmoset MRI structural scan was collected as part of the development of the NIH marmoset brain atlas (*Liu et al., 2018*; *Liu et al., 2020*). Data was collected ex vivo from a 4.5-year-old male marmoset using a T2-star weighted 3D FLASH sequence using a horizontal MRI scanner (Biospec 7 Tesla, Bruker Biospin, Ettlingen, Germany).

A total of 10 scans were collected and averaged into one final image. In combination with scans from other modalities, cortical boundaries were manually delineated on each coronal slice. Boundaries were then refined through comparisons with other atlases. Volumetric data was then converted to surfaces using a custom pipeline involving an intermediate generation of high-resolution mesh data

(*Madan, 2015*), decimation (*Madan, 2016*), and remeshing. Reconstructed pial and white matter surfaces were visually quality controlled in conjunction with the MR images.

## Other non-human primates (NHPs)

The remaining NHP MRIs and subsequent brain surface extraction are detailed in *Ardesch et al., 2022*; *Bryant et al., 2021*, and provided to the authors in a processed format. Briefly, a range of specialised scanners were used to acquire optimal images for each species. FreeSurfer 6.0 with some modifications was used for surface reconstruction, complemented by FSL, ANTS, and Matlab. All surfaces were visually inspected for accuracy and consistency across datasets.

## Comparative neuroanatomy data

The comparative neuroanatomy dataset for different mammalian species is the same as previously published (*Mota and Herculano-Houzel, 2015*). Note that for this dataset, we only had numerical values for the total and exposed surface area, as well a average cortical thickness estimates. We did not perform any analysis across scales in this dataset (hence surfaces were not required), but only used it as a reference dataset.

### Ethics statement

All analyses were performed on anonymised data that were acquired previously as part of other studies/consortia with ethical approval from Newcastle University (reference: 22/SC/0016).

### Statistical analyses

Briefly, linear regression is used in either a mixed-effect model to capture effects across individuals and species, or in simple fixed-effect settings to estimate regression slopes to obtain fractal dimension.

In the final part of the Results, we analyse the effect between a group of 20-year-olds and 80-year-olds. Effect size is calculated as Cohen's D between the two groups.

Throughout the paper, statistical significance is not a crucial argument, and we report p-values only for reference and completeness.

More details can be seen in the analysis code, from which the reader can directly reproduce all the main result figures.

## Results

### All primate brains follow the same scaling law across spatial scales

We have analysed cortices of 11 different primate species: Gray-bellied Night Monkey (*Aotus lemurinus*), Tufted Capuchin Monkey (*Cebus apella*), Black-and-white Colobus (*Colobus guereza*), Senegal Bushbaby (*Galago senegalensis*), Woolly Monkey (*Lagothrix lagotricha*), Gray-cheeked Mangabey (*Lophocebus albigena*), Rhesus Macaque (*Macaca mulatta*), Common Marmoset (*Callithrix jacchus*), Chimpanzee (*Pan troglodytes*), White-faced Saki (*Pithecia pithecia*), and various cohorts of human subjects. We applied our coarse-graining procedure to their pial and white matter surfaces, and empirically determined (i) that all species followed a power law (linear regression $R^2 > 0.999$ for all species); (ii) the slope of said power law is $\alpha = 1.255$ on a group level (CI: [1.254 1.256]) using linear mixed effect modelling (see *Figure 2* for visualisation, and Appendix 3 for a detailed breakdown by species); and importantly, (iii) all species also show a similar offset $\log k \approx -0.65263$, with a standard deviation of intercept across species estimated at 0.02 from linear mixed effect modelling in $\log k$.

Taken separately, the scaling for each species is proof that their cortices are self-similar with the same scaling: they each approximate a fractal with fractal dimension $d_f = 2.5$. Considering all species together, different species also overlap substantially (similar offset), and only differ from each other in the range of scales over which the approximation is valid (see Appendix 3.2 and Appendix 4). Thus, as *Figure 2* illustrates, the data supports a *universal* scaling law across primate species and spatial scales:

$$A_t(\lambda)T(\lambda)^{\frac{1}{2}} = kA_e(\lambda)^{\frac{5}{4}}, \tag{3}$$

with $k = 0.2277$.

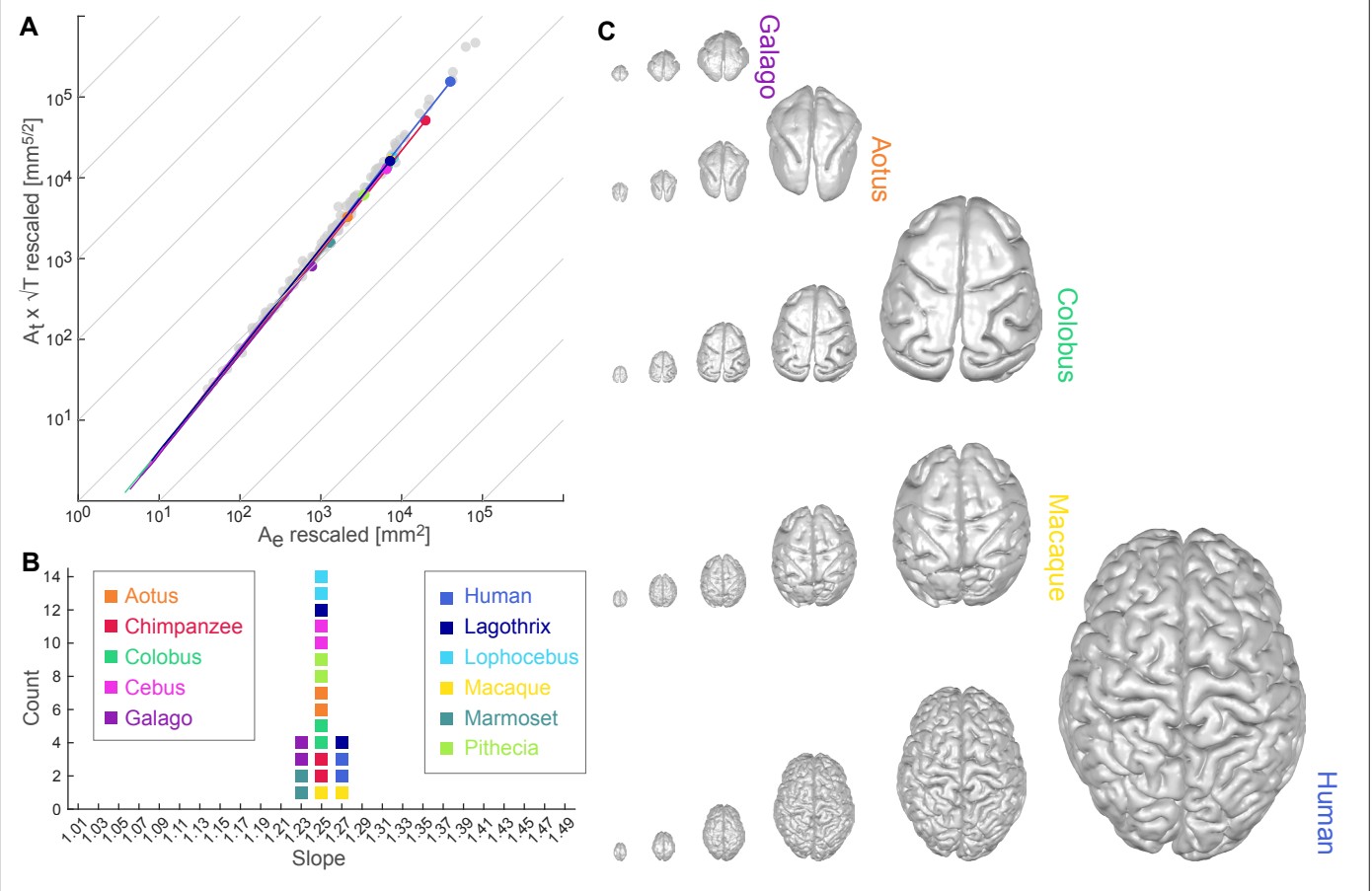

**Figure 2.** Universal scaling law for 11 coarse-grained primate brains. (**A**) Coarse-grained primate brains are shown in terms of their relationship between $\log_{10}(A_t \times \sqrt{T})$ *vs.* $\log_{10}(A_e)$. Each solid line indicates a cortical hemisphere from a primate species. Thin grey lines indicate a slope of 1 for reference. Filled circles mark the data points of the original cortical surfaces. Grey data points are plotted for reference and show the comparative neuroanatomy dataset across a range of mammalian brains (***Mota and Herculano-Houzel, 2015***). (**B**) Slopes ($\alpha$) of the regression of data points in A for each species. For each species, two data points are shown, one per hemisphere. Colour-code for each species is maintained throughout the whole figure. (**C**) Rescaled brain surfaces visualised for five example species at different levels of coarse-graining.

## Primate brains at different spatial scales are morphometrically similar to each other and other mammalian species

To better characterise the coarse-grained cortices in terms of their similarity in offset, we use a set of independent morphometric measures, $K$, $I$, and $S$, that summarise the morphometry of the cortex in a natural and statistically robust way. We can, therefore, assess the offset $K$ (and shape term $S$) without interference by the isometric size or rescaling.

We can measure $K$ and $S$ for any object, but a fuller expression is captured by the *trajectory* of said object as a function of coarse-graining in the $K \times S$ plane. This is a very convenient and informative way of summarising an object: self-similar objects correspond to straight trajectories as the $K \times S$ plane is in log-log space. In particular, objects without any folds or protrusion (i.e. convex, such as the box with finite thickness in ***Figure 3A***) corresponds to the line $K = -\frac{1}{9}S$, as $A_e = A_t$ for all levels of coarse-graining. Horizontal trajectories (constant $K$) represent fractal objects with fractal dimension $d_f = 2.5$ (***Figure 3B***). And finally, in the $K \times S$ plane, a group of objects can said to be 'universal' when their trajectories overlap, so that they can all be regarded as coarse-grained versions of one another (***Figure 3C***).

Primate cortices (***Figure 3D***) display a nearly invariant $K$ in all cases. But, over all levels of coarse-graining, $K$ also remains near-invariant in all *trajectories* as $S$ decreases, resulting in a set of horizontal lines that largely overlap with each other and other mammalian species. The variance in $K$ across scales and all 11 species is $< 0.01$, which is at least an order of magnitude lower than the variance in

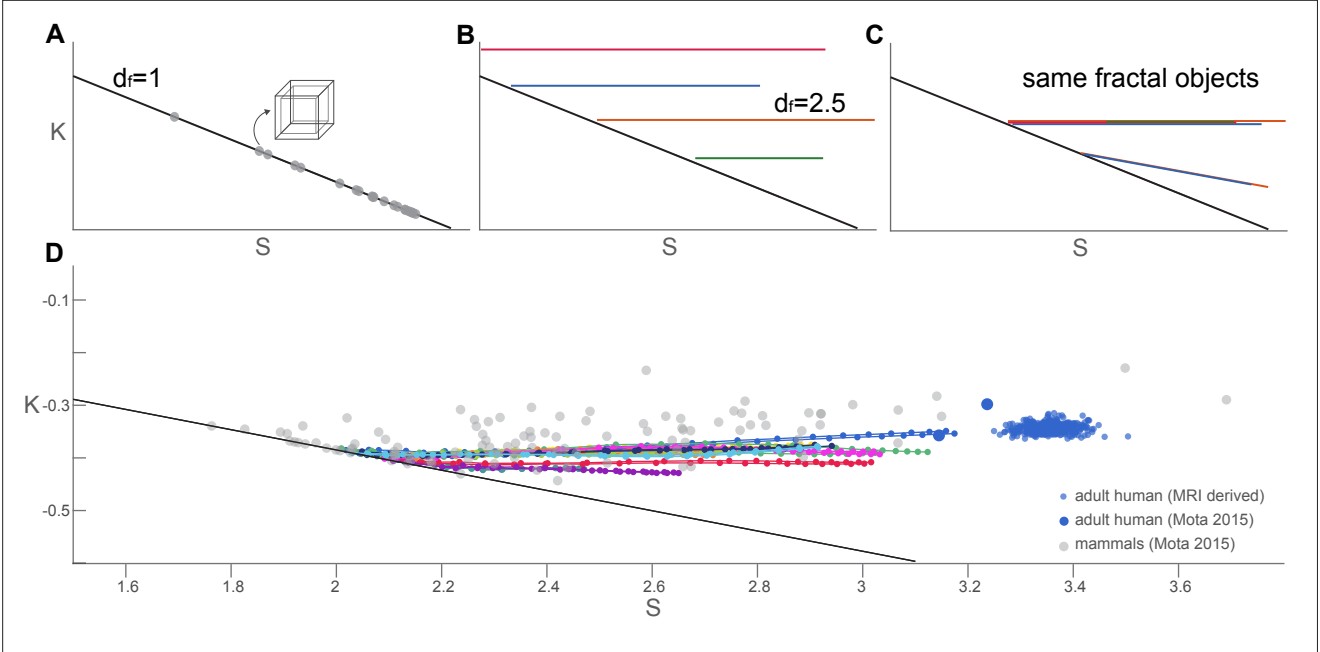

**Figure 3.** Trajectories of coarse-grained primate cortices and other mammalian and human brains in $K \times S$ plane. (**A**) Straight trajectories indicate self-similarity (described by a scaling law). In particular, the black line here indicates objects with $A_e = A_t$ for all scales, such as the box of finite thickness with a fractal dimension $d_f = 1$ (grey data points). This line is reproduced in all subpanels for reference. (**B**) Morphological trajectories of multiple hypothetical fractal objects are shown which. A flat trajectory (constant $K$) corresponds to $d_f = 2.5$ in this space. However, these objects are clearly different fractals with different values of $K$. (**C**) Hypothetical objects with overlapping straight morphological trajectories indicate multiple realisations of the same fractal object. Flat trajectories (constant $K$) correspond to $d_f = 2.5$. The two hypothetical objects with a decreasing $K(S)$ correspond to $2 < d_f < 2.5$ (**D**) Projecting our actual data into the normalised $K \times S$ plane showing the coarse-grained primate brains (same as in **Figure 2**) as data point connected with solid lines (colour-code same as **Figure 2**). Different mammalian brains are shown as grey scatter points, and adult human data points are blue.

$S$. Primate brains, therefore, have all three characteristics of self-similarity, fractality (with $d_f = 2.5$), and universality (invariant $K$ for all scales and species) at the same time.

Thus, coarse-grained primate cortices are morphometrically similar to, and in terms of the universal law, 'as valid as' actual existing mammalian cortices. Note, of course, that the coarse-grained brain surfaces are an output of our algorithm alone and not to be directly/naively likened to actual brain surfaces, e.g., in terms of the location or shape of the folds. Our comparisons here between coarse-grained brains and actual brains is purely on the level of morphometrics across the whole cortex. In contrast, we tested various non-brain objects, and while e.g., the walnut, and bell pepper form (partially) straight lines, they vary in both K and S (see Appendix 5). These objects may have a fractal regime, but their fractal dimension is not 2.5, nor are they similar to primate or mammalian brains in terms of $K$ or $S$. Furthermore, Appendix 6 underscores the algorithmic and statistical robustness of these results using multiple realisations of the coarse-graining procedure on the same object. In this framework, our main result can thus be expressed simply: for all the cortices we analysed, and for none of the non-cortices, coarse-graining will leave $K$ largely unaffected, while morphological complexity $S$ will decrease.

## Morphometric measures as functions of scale reveal scale-specific effects of ageing

In the final part of our work, we show how our algorithm and the associated new understanding of brain morphology may become useful in applications. As an example, we will focus on how the ageing process affects human cortical morphology across scales. In **Figure 4A**, we compare the total surface area $A_t(\lambda)$ as a function of scale $\lambda$ for a young (20- year-old) *vs.* an old (80- year-old) group of human brains. The difference in $A_t$ between the groups takes a U shape, and the strongest effect is seen at approximately 2 millimeters (greatest effect size of –8.635 seen at scale 2.188 mm), where older

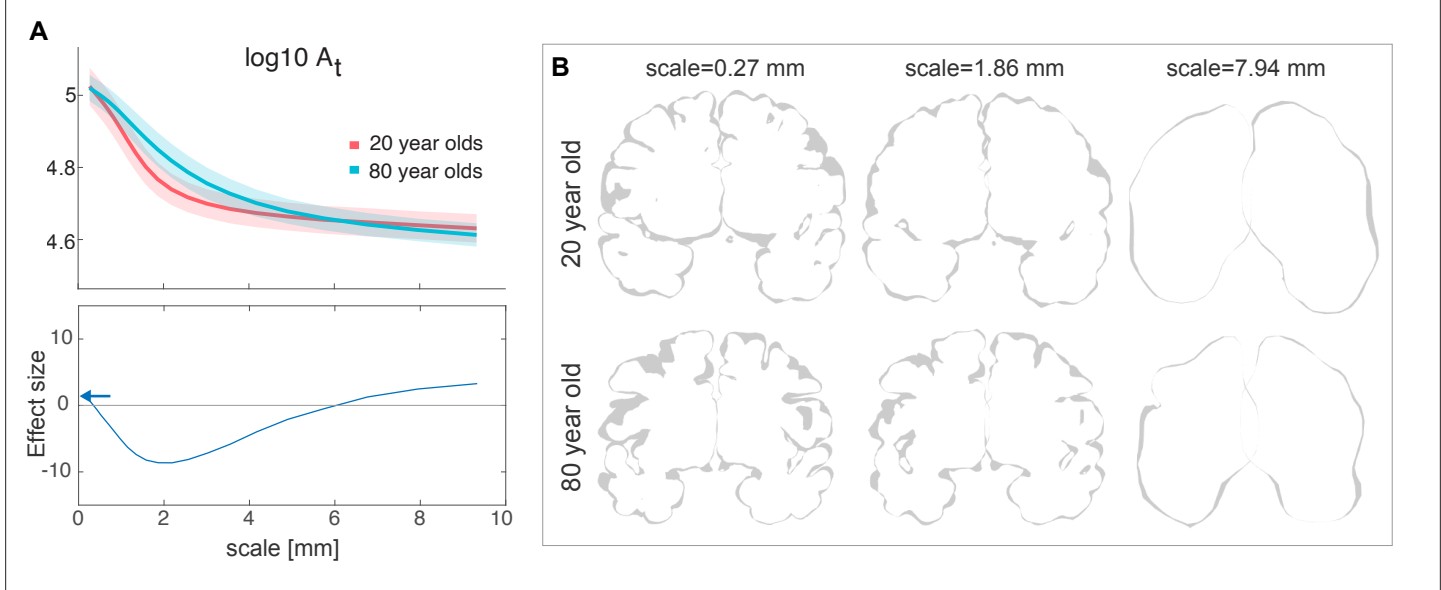

**Figure 4.** Human ageing shows differential effects depending on spatial scale. (**A**) Top: $A_t(\lambda)$ is shown for a group of 20-year-olds (red, n=27) and a group of 80-year-olds (blue, n=86). Mean and standard deviation are shown as the solid line and the shaded area, respectively. Bottom: Effect size (measured as rank sum z-values) between the older and younger groups at each scale. Positive effect indicates a larger value for the younger group. Blue arrows indicate the effect size at the 'native scale' i.e., using the original free surfer grey and white matter meshes. (**B**) Coronal slices of the pial surface of an example 20-year-old subject and an 80-year-old subject at different scales (columns).

subjects have higher $A_t$. For scales over ~5 mm, and under ~0.5 mm the differences become relatively small, suggesting the ageing process has less effect on the largest and smallest cortical morphological features. Finally, we reproduced these results in an independent dataset in Appendix 7.2.

In this particular example, the scale-dependency of morphological measures can be visually and intuitively understood by looking at the reconstructed surfaces at each scale: *Figure 4B* shows some coronal slices of the cortical surface. On a scale of 0.27 mm, the gyri in the younger subjects are densely packed, but the older subjects show the expected widening between gyral walls and decrease in gyral surface area at the crown (see e.g. *Jin et al., 2018*; *Madan, 2019*; *Kochunov et al., 2008* for recent investigations and references therein). At 1.86 mm, the younger cortices have already partially 'melted,' erasing most small sulci between the densely packed gyri. In the older humans, however, the gyri are less dense, the sulci is more open, and thus, at 1.86 mm, most gyri and sulci have not been erased yet. At scale 7.94 mm, both young and old brains have 'melted' down to similarly near-lissencephalic cortices. For a more detailed multiscale investigation over the entire human lifespan, please refer to our new preprint (*Leiberg et al., 2024*).

More broadly, one can regard the melting process as a way of determining how cortical area is allocated across different scales. As the cortex 'melts,' the contributions to the total area from features smaller than the cut-off scale are eliminated. In the example in *Figure 4*, for instance, we can say about half ($10^{5-4.7} = 10^{0.3} \approx 2$) of the total area in the 80 y.o. cortices are present in features smaller than 4 mm.

## Discussion

We have devised a new way of expressing the morphology of the mammalian cerebral cortex, as the flow in the values of morphometric measures over a range of spatial scales. This was achieved by coarse-graining cortical surfaces, erasing morphological features smaller than the specified scale, while preserving surface integrity. After applying this method to the cortices of 11 primate species, we have shown that all these diverse cortices are approximations of *the same archetypal self-similar (fractal) shape*. Most of their morphological diversity can be ascribed to the species-specific ranges of spatial scales over which each approximation is valid, with the smallest scale being an invariant multiple of cortical thickness. This was a proof-by-construction of fractality, a step beyond the usual

box-counting approach, which also yields scale-dependent morphometrics. As a proof-of-principle we showed that healthy human ageing has highly scale-dependent effects in a range of morphometrics.

## Advantages and advances

Compared to previous literature, we can summarise our main contribution and advance as follows: (i) We are showing for the first time that representative primate species follow the exact same fractal scaling – as opposed to previous work showing that they have a similar fractal dimension (*Hofman, 1985*; *Hofman, 1991*), i.e., slope, but not necessarily the same offset, as previous methods had no consistent way of comparing offsets. (ii) Previous work could also not show direct agreement in morphometrics between the coarse-grained brains of primate species and other non-primate mammalian species. (iii) Demonstrating in proof-of-principle that multiscale morphometrics, in practice, can have much larger effect sizes for classification applications. This moves beyond our previous work where we only showed the scaling law across (*Mota and Herculano-Houzel, 2014*) and within species (*Wang et al., 2016*), but all on one (native) scale with comparable effect sizes for classification applications (*Wang et al., 2016*).

In simple terms: we know that objects can have the same fractal dimension, but differ greatly in a range of other shape properties. However, we demonstrate here, that representative primate brains and mammalian brains indeed share a range of other key shape properties, on top of agreeing in fractal dimension. This suggests a universal blueprint for mammalian brain shape and a common set of mechanisms governing cortical folding. As a practical additional outcome of our study, we could show that our novel method of deriving multiscale metrics can differentiate subtle morphological changes much better (four times the effect size) than the metrics we have been using so far at a single native scale.

Expressing cortical morphology as a function of scale is more detailed than a list of summary morphometric measures, and more informative than the mere listing of every sulcus and gyrus. We propose this new syntax as the basis for a more rigorous characterisation of brain morphology and morphological changes. A clear advantage is that some biological processes may only act on a specific spatial scales, leaving other scales untouched (ageing in our example). By disambiguation of the spatial scale, it allows for an extra dimension of understanding, and tracking of biological processes. In the future, this approach can also be extended to cortical development and to various degenerative (*Wang et al., 2016*) and congenital (*Wang et al., 2021*) neuropathic conditions, especially if combined with a regionalised versions of this method applied to specific cortical regions (*Wang et al., 2019*) or local patches (*Leiberg et al., 2021*).

## Implications of universality

Empirically, the main result of this paper is the demonstration of a universal self-similar scaling (*Equation 3*) for primate, and presumably mammalian cortices. There are two aspects of this universality: first that for each and every cortex the value of $K$ remains the same for all scales as one removes substructures smaller than a varying length scale (or equivalently, that the fractal dimension is almost exactly 2.5 in all cases). Second, that the value of $K$ for the cortices of different species is approximately the same, as previously observed (*Equation 1*) across species (*Mota and Herculano-Houzel, 2015*) and individuals (*Wang et al., 2016*). One could imagine a set of objects for which one but not the other aspect of the universality in $K$ holds true. However, the fact that both universalities hold true is significant. It suggests the existence of a single highly conserved mechanism for cortical folding, operating on all length scales self-similarly with only a few morphological degrees of freedom. It also hints at the possibility of deriving cortical scaling from some variational principle. Finally, this dual universality is also a more stringent test for existing and future models of cortical gyrification mechanisms at relevant scales, and one that moreover is applicable to individual cortices. For example, any models that explicitly simulate a cortical surface as an output could be directly coarse-grained with our method and the morphological trajectories can be compared with those of actual humans and primate cortices. The simulated cortices would only be 'valid' in terms of the dual universality, if it also produces the same morphological trajectories (Note, we do not suggest to directly compare coarse-grained brain surfaces with actual biological brain surfaces. As we noted earlier, the coarse-grained brain surfaces are an output of our algorithm alone and are not to be directly/naively likened to actual brain surfaces, e.g., in terms of the location or shape of the folds. Our comparisons here

between coarse-grained brains and actual brains is purely on the level of morphometrics across the whole cortex).

The scaling itself does not imply or favour any particular proposed gyrification model (ours *Mota and Herculano-Houzel, 2015* included), and all results in this paper are agnostic about this choice. Indeed, our previously proposed model (*Mota and Herculano-Houzel, 2015*) for cortical gyrification is very simple, assuming only a self-avoiding cortex of finite thickness experiencing pressures (e.g. exerted by white matter pulling, or by CSF pressure). The offset $K$, or 'tension term,' precisely relates to these pressures, leading us to speculate that subtle changes in $K$ correlate with changes in white matter property (*Wang et al., 2016*; *Wang et al., 2021*). In the same vein of speculation, the scale-dependence of $K$ shown in this work might, therefore, be related to different types of white matter that span different length scales, such as superficial *vs.* deep white matter, or U-fibres *vs.* major tracts. However, there are also challenges to the axonal tension hypothesis (*Xu et al., 2010*). Indeed, white matter tension differentials in the developed brain may not explain the location of folds, but instead white matter tension may contribute to a whole-brain scale 'pressure' during development that drives the folding process overall. Aside from speculations about the biological interpretation, the simplicity of the highlighted scaling law parallels many complex phenomena in nature that displays simple and universal scaling that can be derived from first principles (*Barenblatt, 1996*; *West et al., 1997*; *Gagler et al., 2022*). In addition, recent results suggest simplicity and symmetry are generically favoured on statistical-ensemble grounds by evolution (*Johnston et al., 2022*). Our model correctly predicts the scaling law (*Equation 1*), but a more complete explanation for cortical gyrification is probably far more complex (*Quezada et al., 2020*) than can be accounted by such a simple model.

One specific example of said complexity is the exact patterns, locations, depth, and features of gyri and sulci. We know such patterns to be, for example, variable but also somewhat heritable in humans, whilst in macaques such patterns are relatively preserved across the species. Our work does not explain any of these observations, nor are the coarse-grained versions of human brains is supposed to exactly resemble the location/pattern/features of gyri and sulci of other primates. The similarities we highlighted here are on the level of summary metrics, and our goal was to highlight the universality in such metrics points towards highly conserved quantities and mechanisms.

## Biological plausibility and implications

The observation that with increasing voxel sizes, the coarse-grained cortices tend to be smoother and thicker is particularly interesting: the scaling law in *Equation 3* can be understood as thicker cortices ($T$) form larger folds (or are smoother i.e. with less surface area $A_t$) when brain size is kept constant ($A_e$). This way of understanding has also been vividly illustrated by using the analogy of forming paper balls with papers of varying thickness in *Mota and Herculano-Houzel, 2014*: comparing two paper balls of the same size ($A_e$) will show that the one that uses thicker paper ($T$) will be smoother, have larger folds and a smaller total surface area ($A_t$), in comparison with the one using thinner paper. The scaling law can, therefore, be understood as a physically and biologically plausible statements and our algorithm yields results in line with the scaling law.

More broadly, the interaction between brain development and evolution may also benefit from a scale-specific understanding. This may be important in elucidating what in cortical morphology is selected for by evolution, what is determined by physics; what is specified by genes, and what is emergent. For example, one can estimate the number of structural features at each scale ($A_t(\lambda)$ as multiples of $A_0(\lambda)$), and it will be interesting to correlate this number to other quantifiers of cortical structure, such as the number of neurons (*Mota and Herculano-Houzel, 2014*), the number of functional areas (*Molnár et al., 2014*), or the number of cortical columns (*Kaas, 2012*), possibly over different stages of development. Generally, the larger the cortical feature (i.e. from gyri to functional areas to lobes to hemispheres), the earlier during development it appears (*Zilles et al., 2013*; *Garcia et al., 2018*), and more broadly it is conserved over kinship (*Pizzagalli et al., 2020*) and phylogeny (*Heuer et al., 2019*; *Valk et al., 2020*). It thus seems likely that comparative neuroanatomical methods (*Mars et al., 2014*; *Croxson et al., 2018*) may be directly used to identify and contrast structures in coarse-grained cortices of more highly gyrified species with their analogues in less-gyrified species. One could then perhaps specify when evolution conserves and when it invents old and new cortical features.

From an application perspective, our final result illustrates clearly that the surface area difference between older and younger subjects at the 'native' scale (i.e. original free surfer surfaces) is negligible

(effect size smaller than two standard deviations). However, in our analysis across scales, there is a clear optimal scale at ~2 mm where the effect size is maximised between older and younger subjects (effect size is –8 standard deviations). For most classification applications in biology and medicine, the increased effect size and hence separability of groups in the scale-dependent morphometrics represent a huge advance over the native scale.

## Outlook

Our work here was limited to summary descriptors of entire cortical hemispheres, but future work will explore extensions of these methods to lobes and cortical areas, similarly to *Wang et al., 2019*; *Leiberg et al., 2021*. This will generate precise characterisations of the morphological differences between phylae and across developmental stages, and perhaps pinpoint the time and location of morphological changes leading to congenital and neurodegenerative conditions. Ultimately, we hope this new framework for expressing and analysing cortical morphology, besides revealing a hitherto hidden regularity of nature, can become a powerful tool to characterise and compare cortices of different species and individuals, across development and ageing, and across health and disease.

## Acknowledgements

We thank members of the Computational Neurology, Neuroscience, and Psychiatry Lab (https://www.cnnp-lab.com) for discussions on the analysis and manuscript, and Dirk Jan Ardesch and Martijn van den Heuvel for helpful discussions and NHP brain surface data. PNT and YW are both supported by UKRI Future Leaders Fellowships (MR/T04294X/1, MR/V026569/1); YW and KL are further supported by the EPSRC (EP/Y016009/1, EP/L015358/1). B Mota is supported by Fundação Serrapilheira Institute (grant Serra-1709–16981) and CNPq (PQ 2017 312837/2017-8).

## Additional information

### Funding

| Funder | Grant reference number | Author |
|---|---|---|
| Engineering and Physical Sciences Research Council | EP/L015358/1 | Yujiang Wang Karoline Leiberg |
| Engineering and Physical Sciences Research Council | EP/Y016009/1 | Karoline Leiberg |
| UK Research and Innovation | MR/V026569/1 | Yujiang Wang |
| UK Research and Innovation | MR/T04294X/1 | Peter Neal Taylor |
| Instituto Serrapilheira | Serra-1709-16981 | Bruno Mota |
| Conselho Nacional de Desenvolvimento Científico e Tecnológico | PQ 2017 312837/2017-8 | Bruno Mota |

The funders had no role in study design, data collection and interpretation, or the decision to submit the work for publication.

### Author contributions

Yujiang Wang, Conceptualization, Resources, Data curation, Software, Formal analysis, Funding acquisition, Validation, Investigation, Visualization, Methodology, Writing - original draft, Project administration, Writing – review and editing; Karoline Leiberg, Software, Validation, Writing – review and editing; Nathan Kindred, Resources; Christopher R Madan, Colline Poirier, Christopher I Petkov, Resources, Writing – review and editing; Peter Neal Taylor, Validation, Visualization, Writing – review and editing; Bruno Mota, Conceptualization, Investigation, Methodology, Writing – review and editing

### Author ORCIDs

Yujiang Wang ⓘ https://orcid.org/0000-0002-4847-6273

Christopher R Madan https://orcid.org/0000-0003-3228-6501
Colline Poirier http://orcid.org/0000-0001-9793-4907
Christopher I Petkov http://orcid.org/0000-0002-4932-7907
Bruno Mota http://orcid.org/0000-0002-6950-4106

Reviewer #2 (Public review): https://doi.org/10.7554/eLife.92080.4.sa1
Reviewer #3 (Public review): https://doi.org/10.7554/eLife.92080.4.sa2
Author response https://doi.org/10.7554/eLife.92080.4.sa3

## Additional files

### Supplementary files
• MDAR checklist

### Data availability

The code for coarse-graining has been integrated into our MATLAB toolbox Cortical Folding Analysis Tools: https://github.com/cnnp-lab/CorticalFoldingAnalysisTools (copy archived at *Wang et al., 2024*), which now also includes a graphical user interface. Users will see the latest updates in this repository. The analysis code underpinning this paper is published on GitHub: https://github.com/cnnp-lab/2024_Folding_scales/ (copy archived at *Wang, 2024*). The post-processing data (i.e. 'voxelisations' and derived metrics) are uploaded on Zenodo: https://doi.org/10.5281/zenodo.12820611. The data, together with the code, will allow readers to reproduce of our main results.

The following dataset was generated:

| Author(s) | Year | Dataset title | Dataset URL | Database and Identifier |
|---|---|---|---|---|
| Wang Y | 2024 | Data underlying "Neuro-evolutionary evidence for a universal fractal primate brain shape" | https://doi.org/10.5281/zenodo.12820611 | Zenodo, 10.5281/zenodo.12820610 |

The following previously published dataset was used:

| Author(s) | Year | Dataset title | Dataset URL | Database and Identifier |
|---|---|---|---|---|
| Ardesch DJ, Scholtens LH, de Lange SC, Roumazeilles L, Khrapitchev AA, Preuss TM, Rilling JK, Mars RB, van den Heuvel MP | 2022 | Primate Brain Bank MRI | https://doi.org/10.5281/zenodo.5044935 | Zenodo, 10.5281/zenodo.5044935 |

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

## Appendix 1

### Coarse-graining algorithm

The core coarse-graining algorithm underpinning our analyses takes pial and white matter surfaces from the cortical surface reconstruction as inputs and at any specified spatial scale $\lambda$ 'voxelises' the grey matter ribbon at the specified resolution. To achieve this, we set up a 3D voxel grid, where each grid voxel is of size $\lambda \times \lambda \times \lambda$.

In detail, we assign all voxels in the grid with at least four corners inside the original pial surface to the pial voxelisation. This process allows the exposed surface to remain approximately constant with increasing voxel sizes. A constant exposed surface is desirable, as we only want to gradually 'melt' and fuse the gyri, but not grow the bounding/exposed surface as well. We want the extrinsic area to remain approximately constant as we decrease the intrinsic area via coarse-graining; it is like generating iterates of a Koch curve in reverse, from more to less detailed, by increasing the length of the smallest line segment.

We then assign voxels with all eight corners inside the original white matter surface to the white matter voxelisation. This is to ensure integrity of the white matter, as otherwise white matter voxels in gyri may become detached from the core white matter, and thus artificially increase white matter surface area. Indeed the main results of the paper are not very sensitive to this decision using all eight corners, vs. e.g., only four corners, as we do not directly use white matter surface area for the scaling law measurements. However, we still maintained this choice in case future work wants to make use of the white matter voxelisations or derivative measures.

Finally, subtracting the white matter voxelisation from the pial voxelisation, we obtain the grey matter voxelisation (*Figure 1B* bottom row). The pial surface voxelisation is designed such that neighbouring gyri can fuse if their separation is smaller than the scale of interest, whilst not growing the cortex outwards. The white matter surface voxelisation allows the walls of each gyrus to thicken and fuse inwards. Visually, this process looks as if the cortex is 'thickening' inwards and smoothing on the outside.

From the voxelised cortex at each scale $\lambda$, we can then obtain coarse-grained pial and white matter surfaces, and extract estimates of global morphometric measures such as the average cortical thickness $T(\lambda)$, the total cortical surface area $A_t(\lambda)$, and the exposed surface area $A_e(\lambda)$. For each $\lambda$, we derived an outer isosurface equal to 0.5 for the pial voxelisation (voxel values are 1 within the voxelised pial surface and 0 otherwise). This isosurface is then defined as the coarse-grained pial surface at this scale. The surface area of this isosurface is used as $A_t(\lambda)$. The exposed surface area $A_e(\lambda)$ is subsequently derived from the convex hull of the pial isosurface. Finally, the cortical thickness $T(\lambda)$ is estimated as $\frac{V_G(\lambda)}{A_t(\lambda)}$, where $V_G(\lambda)$, which is the estimated grey matter volume, derived from the number of grey matter voxels, is multiplied by the voxel volume. These global morphometric measures are summary statistics of the brain at each particular scale, capturing information about both their intrinsic geometry ($A_t(\lambda), T(\lambda)$) and extrinsic geometry ($A_e(\lambda)$). In this manner, for each cortex, we obtain not just one set of summary morphometric measures, but rather a set of measures as functions of $\lambda$. A detailed walk-through of the coarse-graining algorithm and estimation of morphometric measures is provided on Github: https://github.com/cnnp-lab/Cortical FoldingAnalysisTools/blob/master/Scales/fastEstimateScale.m.

Note that although this process is inspired by the box-counting algorithm, it is different from box-counting and related convolution-based algorithms: we do not simply apply successive convolutions with an increasing kernel size (or equivalent), which effectively would achieve a uniform and spatially isometric 'smearing' of the original cortical ribbon to a given scale, rather than a targeted erasure of *surface* details smaller than said scale. As a result, this method yields well-defined and well-behaved white and gray matter surfaces. Thus, one can apply all the usual analytical tools to these realisations that are applicable to actual cortices. Note that our approach is more comparable in its principles to the calculation of the outer smoothed pial surface in FreeSurfer (*Schaer et al., 2008*), which utilises a dilation *and* erosion convolution to effectively erase details below a certain scale. Of course, our proposed procedure is not the only conceivable way to erase morphological details below a given scale; and we are actively working on related algorithms that are also computationally cheaper. Nevertheless, the current version requires no fine-tuning, is computationally feasible and conceptually simple, thus making it a natural choice for introducing the methodology and approach.

Given how broadly it has been verified, we expect the observed universality in cortical self-similar scaling to be robust to the details of the coarse-graining algorithm. It would be very informative to test this proposition in future. For example, an alternative method inspired by *Yu et al., 2021* could be implemented, eschewing voxelisation and dealing only with the flow of nested surfaces with self- and mutual- avoidance explicitly implemented. Going in the opposite direction, more detailed models for the mechanisms of cortical folding (see e.g. *Raznahan et al., 2011*) can be regarded as a type of reverse melting, and could perhaps be implemented and tested in a similar fashion as fine-graining procedures.

## Appendix 2

### Influence of original image resolution

In this section, we want to demonstrate the relatively weak effect of the original image resolution on our analysis outputs. To this end, we used five example HCP subjects, who were scanned at 0.7mm isotropic image resolution, and downsampled their images to 1mm isotropic images. We then proceeded with our analysis using two freesurfer outputs. (1) the HCP freesurfer pipeline output optimised for the 0.7mm resolution, and (2) a standard freesurfer pipeline output on the 1mm downsampled images. We proceeded with these two sets of surfaces in our analysis and show the resulting morphology measures in *Appendix 2—figure 1*.

We observe a relatively weak difference between these two sets of inputs/surfaces. Both sets largely follow the same trajectory across scales. Especially in the exposed area, the within and between subject differences are noticeably larger than between image resolutions. In total area and thickness, some small but systematic differences are seen in all subjects between the scales of 1- 2mm. These are most likely differences in the cortical morphology reconstruction in the freesurfer surfaces using the two different resolution images as input. However, in none of these resulting morphology measures do we see an artifact specifically at 0.7 or 1mm. Any differences between image resolutions only result in subtle changes in the freesurfer meshes, which are smooth. Our analysis method, therefore, has no direct dependency on the image resolution, as our inputs are these smooth freesurfer meshes.

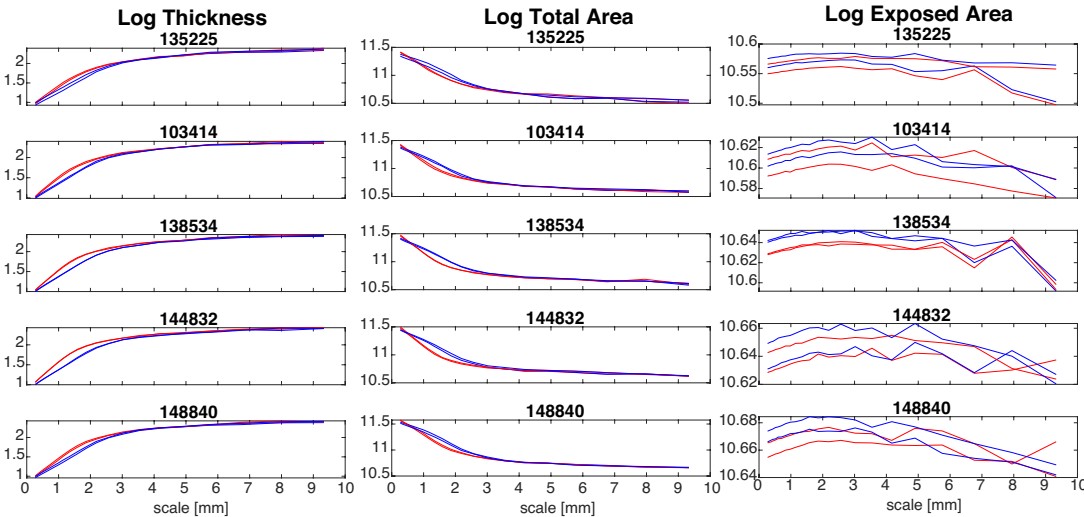

**Appendix 2—figure 1.** Effect of using different original image resolutions in five examples Human Connectome Project (HCP) subjects. In all plots, red lines indicate an original image resolution of 0.7 mm isotropic, whereas blue line indicate an original image resolution of 1 mm isotropic. Both hemispheres are shown for each subject.

## Appendix 3

### Scaling properties by species

#### 3.1 Obtaining the scaling law

From the coarse-grain procedure (*Figure 1*), we obtain surface meshes for the pial and white matter surface at each spatial scale. From those, we derive exposed area, total pial surface area, and average cortical thickness as described in Methods. However, the coarse-graining procedure, by itself, barely changes the exposed surface area, and only changes the total surface area minimally (*Appendix 3—figure 1A*). As the voxel size changes at each scale, it only starts to affect the exposed area at very large scales, where effectively the voxels are no longer a good description of the shape of the skull. Thus, if we scatter the raw data points from the coarse-graining procedure in the plane of the scaling law, there is barely any variance in the data (*Appendix 3—figure 1B*). Even after zooming in, we see an almost vertical line (*Appendix 3—figure 1B*). This is expected and is not evidence for or against our hypothesis.

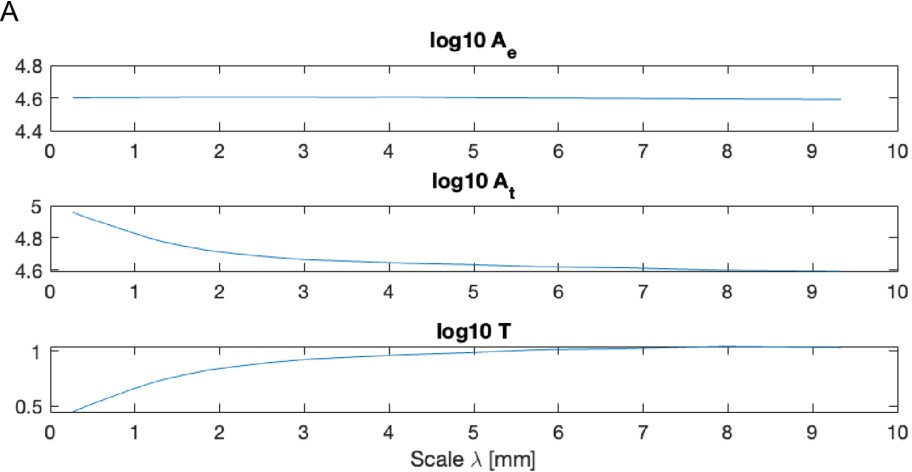

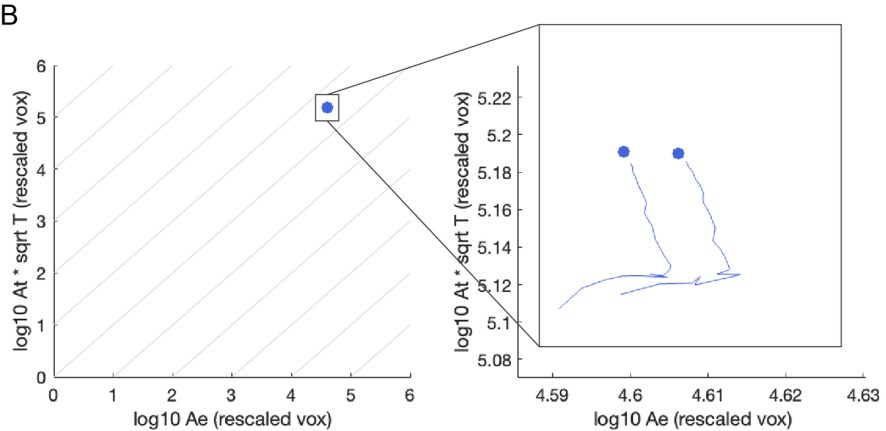

**Appendix 3—figure 1.** Unscaled quantities cannot be used to verify scaling law. (**A**) The exposed surface area $A_e$ barely changes across scales, and $A_t$ only changes minimally. (**B**) Plotted in the scaling law plane (as *Figure 2*) the data points barely have any variance, and overlap each other substantially. Even after zooming in, the trace for the coarse-graining procedure (solid line) is virtually a vertical line (with a small artifactual 'tail' coming from very large voxel sizes).

Instead, we need to transform the data into a perspective that makes sense to be seen in the scaling law plane. Instead of thinking of the coarse-graining procedure as a process that re-renders the cortex at increasing voxel sizes, we can instead think of the procedure as rescaling the original cortical mesh into increasingly smaller sizes, and re-rendering it with the same fixed voxel size. The analogy using Britain's coastline is that instead of measuring the coastline with increasingly smaller

rulers, we resize the map of the coastline to increasingly smaller sizes, but keep the size of the ruler the same. Both procedures are equivalent and produce the same fractal dimension.

To achieve this procedure, we re-scaled our measurements of $A_e$, $T$, and $A_t$ by the voxel size ($\lambda$) and a fixed factor ($l_r$):

$$
\begin{aligned}
A_e^r &= \frac{A_e}{(\lambda \times l_r)^2} \\
A_t^r &= \frac{A_t}{(\lambda \times l_r)^2} \\
T^r &= \frac{T}{(\lambda \times l_r)}.
\end{aligned}
\tag{4}
$$

As we used isometric cubes as voxels, the voxel size refers to the length of a single side. E.g. $\lambda = 1$ if we used a isometric $1 \times 1 \times 1 mm^3$ voxel.

$l_r$ is a fixed factor for each cortical hemisphere, and does not change with $\lambda$. We use it to systematically shift all the data points within a range, such that the re-scaled quantities are not larger than those from the original cortical meshes. One can easily verify that $l_r$ will not change the slope or offset of any scaling law, but simply represents a constant shift to all data points. In our data, some of the re-scaled quantities would indeed be larger than those from the original cortical meshes, as the voxel size we choose is limited at the smaller end only by computational resources. In other words, we can use very small voxel sizes (relative to the mesh), which after re-sizing would yield very large values of $A_e^r$, $A_t^r$, and $T^r$. To avoid this, we chose $l_r$ simply as the ratio of the $I$ (isometric term) of the mesh at the smallest scale we used relative to the original mesh, divided by the $\lambda$ of the smallest scale:

$$
l_r = \frac{1}{\lambda_s} \times \frac{I_s}{I_o},
\tag{5}
$$

where $\lambda_s$ is the smallest voxel size used for a particular cortex, $I_s$ is the corresponding $I$ for this cortex at the smallest voxel size.

Finally, $I_o$ is the $I$ term for the original cortical mesh. Indeed, the ratio of $\frac{I_s}{I_o}$ is always close to, but larger than one in our dataset. Thus, we can see in **Figure 2A** that most traces start very close to the original data point, indicating that our finest scale is reconstructing the original surfaces well.

Note that the re-scaling is isometric, meaning that it only affects $I$, but does not change the data in the $K \times S$ plane (**Figure 3**). Rescaling by a factor proportional to $\lambda^2$ can, therefore, be understood as distributing the data points along the $I$ axis, while $l_r$ can be understood as fixing the position in the $I$ axis relative to the $I$ of the original mesh.

**Figure 2A** is produced with these rescaled quantities as described above. The only final step in producing **Figure 2A**, and in calculating the associated slopes, is the removal of artifactual data points where $\frac{A_t}{A_e} < 0$, which can occur at very large voxel sizes relative to the cortical mesh. The algorithmic implementation in MATLAB can be found on Github: https://github.com/cnnp-lab/CorticalFoldingAnalysisTools/blob/master/Scales/fastEstimateScale.m, as part of our Cortical Folding Analysis MATLAB package https://github.com/cnnp-lab/CorticalFoldingAnalysisTools/, which also has been recently updated with a graphical user interface.

## 3.2 Species-specific details

In the following, we will show the detailed data for each species in terms of their scaling behaviour in **Appendix 3—figure 2**. Videos for all species, showing the pial surface at each scale can be found under https://bit.ly/3CDoqZQ for review purposes. Final versions of all underlying data, analysis code, and videos will be published on Zenodo and eLife upon acceptance of the paper.

Given the demonstrated overlap between species, and if their cortices are all approximations of the same form, how can one tell apart their cortices? The answer is that all approximations have a range of validity, which varies between cortices. For a gyrified cortex of area $A_t$, (When no dependence on $\lambda$ is indicated then we are referring to the values for the original cortex), the coarse-graining will remove details of ever-increasing scale, lowering $A_t(\lambda)$ until attaining lissencephaly for $A_e(\lambda_{lys}) = A_t(\lambda_{lys})$. Indeed, it follows from rewriting **Equation 3** in the form of **Equation 2** that a given cortex' shape will be comprised of self-similar structures with areas ranging from $A_0 = g^{-5} A_t$ at their smallest to $A_t$ at their largest, where $g$ is the gyrification index $g = \frac{A_t}{A_e}$. Whenever this self-similar

scaling is valid, $A_0$ acquires a further interpretation as the typical size of the smallest structures in a cortex: patches smaller than that must be approximately smooth. Consequently, $N_{structures} \simeq \frac{A_t}{A_0} = g^5$ estimates the number of morphological features in each cortex (and adds a new interpretation for the gyrification index). For example, we estimate the human in our dataset to have about 105 morphological features in each cortical hemisphere, and the galago to have about four such morphological features. The corresponding $N_{structures}(\lambda)$ estimates how this number changes over coarse-graining. Appendix 4 later provides a more detailed discussion on this topic.

$\simeq \frac{A_t}{A_0} = g^5$

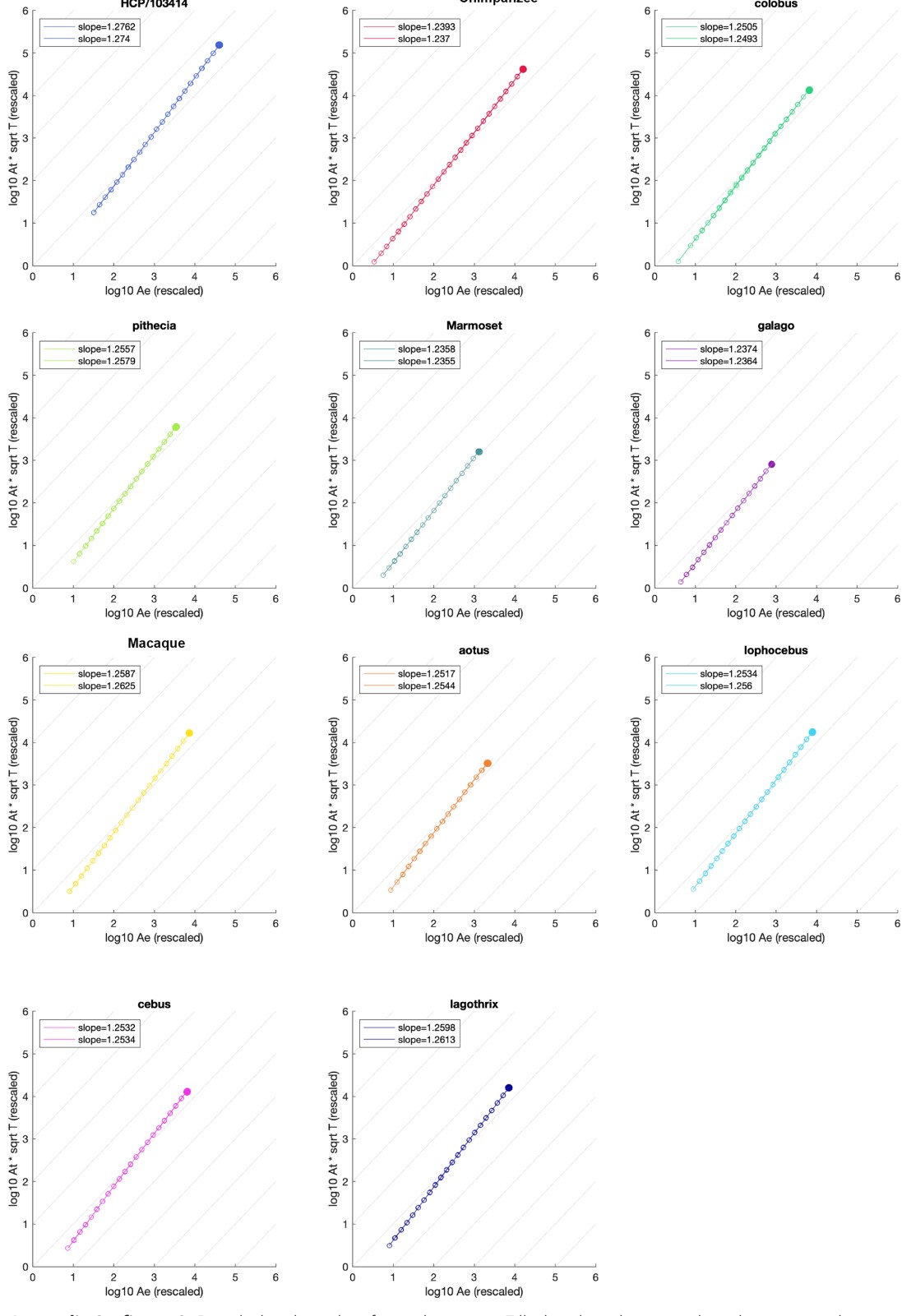

**Appendix 3—figure 2.** Detailed scaling plots for each species. Filled circle is the original mesh, empty circle are data points from the coarse-graining algorithm. Empty circles are connected by a line for visualisation. Two hemispheres are analysed and separated for each species, although data points overlap substantially in plot. Slope estimates are given at top left corner in each plot.

## Appendix 4

### Morphometric relations and the range of validity of the fractal approximation

There are many geometrical quantities that describe aspects of entire cortical hemispheres. But most information contained in such summary morphometric measures is captured, exactly or to good approximations, by the three used in *Equation 1*: the pial (or total) surface area $A_t$, exposed surface area $A_e$, and average cortical thickness $T$. Other quantities, like the gyrification index $g = \frac{A_t}{A_e}$, or the Grey Matter and Total volumes, can be expressed as products of power laws of these quantities; and the logarithm of products of power laws are linear combinations of the logarithms of the constituting variables, with their exponents as coefficients. Thus, the morphology of a given cortex can be fairly summarised as a point in a three-dimensional morphometric space with components given by $\log A_t$, $\log A_e$, and $\log T^2$ (the last exponent guarantees all axes have the same dimension of $\log[Area]$).

This logarithmic structure guarantees that the quantities denoted by products of power laws of the original variables correspond to vectors in the morphometric space: the increase and decrease of their values occur along these vectors, and the planes perpendicular to each vector denote all configurations with the same value for the associated variable. For example, $G = \log g = \log A_t - \log A_e$, the point in the morphometric space associated with the gyrification ratio value $\frac{A_t}{A_e}$, defines a displacement along a vector with coefficients $\{1, -1, 0\}$.

The quantities $K$, $S$ and $I$ used in Sec. 3.2 likewise corresponds to displacements along specific directions in morphometric space, which are all orthogonal to one another. These can be normalised, so that the sum of their squared coefficients equals unity. In this case, expressing cortical shape in the morphometric space in terms of the original variables or the new variables amounts simply to a rotation: a change of orthonormal base.

*Figures 2 and 3* can be then regarded as particular 2D snapshots, taken from a certain direction, of a 3D set of points.

Of particular note, those vectors with a sum of coefficients equal to zero correspond to dimensionless variables. Any such vector is perpendicular to the $I$ direction, and lie on the $K \times S$ plane.

In this framework, we can picture the step-wise coarse-graining of a cortex, as described in Sec. 3.1, as a trajectory in this morphometric space. Empirically, Sec. 3.1, as codified by *Equation 3*, finds that the trajectories in morphometric space for all studied primates are linear, and largely overlap along a line of constant $K$. For cortical morphology, the first fact implies scale-invariance, the second implies universality.

Aging, development, and the progression of neurodegenerative conditions likewise each correspond to different morphometric trajectories.

Now consider a morphometric trajectory, corresponding to the coarse-graining of a cortex up to the point of lissencephaly, i.e., to the spatial scale $\lambda_{lys}$ such that $A_t(\lambda_{lys}) = A_e(\lambda_{lys})$. In the idealised case (but very close to empirical, as seen in Sec. 3.1), all points along the trajectory should follow *Equation 3*, starting at the original values for the morphometric measures and ending at the intersection of the $K = constant$ coarse-graining trajectory and the $g = 1$ line that defines the limit for lissencephaly (points with $g < 1$ are geometrically possible, but are largely avoided by the trajectories of actual cortices, although not by those of all objects; see *Appendix 5—figure 1*). This trajectory spans a range of values for each morphometric measure: for dimensionless morphometric measures, such span is fully specified by *Equation 3*. For all others, one needs also to specify the voxel rescaling.

The scheme described in Sec. Appendix 3.1 simply guarantees that, for a constant $k = A_t A_e^{-\frac{5}{4}} T^{\frac{1}{2}}$, the fundamental area element $A_0 = \frac{T^2}{k^4} = \frac{A_e^5}{A_t^4}$ (and thus also the average thickness $T$) is kept constant for all realisations of the coarse-graining process. We can regard these as approximations of the same original cortex, all drawn with the same resolution, but in ever smaller isometric sizes.

This choice of spatial scale then allows our identification of $A_0$ as the typical size of the smallest morphological features in a given cortex. Equivalently, it is also the area of the largest possible lissencephalic cortex for a given average cortical thickness, just at the cusp of the onset of gyrencephaly.

This same choice also enable us to easily compute the expected span of the various morphometric measures over coarse-graining, assuming the universal scaling: each will range between its original

value (expressed as a function of the original values for $A_t$, $A_e$, and $T$), and its value at spatial scale $\lambda_{lys}$, where $A_t(\lambda_{lys}) = A_e(\lambda_{lys}) = A_0$: we simply replace $A_t$ and $A_e$ by $A_0$, and $T$ by $\sqrt{A_0 k^4}$. The ratios between the initial and final values of all morphometric measures will be given as powers of $g$, imbuing the gyrification index with a new significance.

We list below the expanse for all such morphometric measures ($S = \log s$, $K = \log k$ and $I = \log v_I$)

| Morphometric measure | Original value = | Span $\times$ | Value at lissencephalic limit |
|---|---|---|---|
| $T^2$ | $T^2$ | 1 | $A_0 k^4$ |
| $A_0$ | $\dfrac{A_e^5}{A_t^4}$ | 1 | $A_0$ |
| $A_e$ | $A_e$ | $g^4$ | $A_0$ |
| $A_t$ | $A_t$ | $g^5$ | $A_0$ |
| $V_{total}$ | $\approx \dfrac{2}{9\sqrt{3\pi}} A_e^{\frac{3}{2}}$ | $g^6$ | $\approx \dfrac{2}{9\sqrt{3\pi}} A_0^{\frac{3}{2}}$ |
| $V_{GM}$ | $A_t T$ | $g^5$ | $A_0^{\frac{3}{2}} k^2$ |
| $g$ | $\dfrac{A_t}{A_e}$ | $g$ | 1 |
| $N_{features}$ | $\approx \dfrac{A_t^5}{A_e^5}$ | $g^5$ | 1 |
| $k$ | $A_t A_e^{-\frac{5}{4}} T^{\frac{1}{2}}$ | 1 | $k$ |
| $s$ | $A_t^{\frac{3}{2}} A_e^{\frac{3}{4}} T^{-\frac{9}{2}}$ | $g^{\frac{21}{2}}$ | $k^{-9}$ |
| $v_I$ | $A_t A_e T^2$ | $g^9$ | $A_0^3 k^4$ |
| $\hat{k}$ | $A_t^{\frac{4}{\sqrt{42}}} A_e^{-\frac{5}{\sqrt{42}}} T^{\frac{2}{\sqrt{42}}}$ | 1 | $\hat{k}$ |
| $\hat{s}$ | $A_t^{\frac{2}{\sqrt{14}}} A_e^{\frac{1}{\sqrt{14}}} T^{-\frac{6}{\sqrt{14}}}$ | $g^{\frac{18}{\sqrt{14}}}$ | $\hat{k}^{-3\sqrt{3}}$ |
| $\hat{v}_I$ | $A_t^{\frac{1}{\sqrt{3}}} A_e^{\frac{1}{\sqrt{3}}} T^{\frac{2}{\sqrt{3}}}$ | $g^{3\sqrt{3}}$ | $A_0 \hat{k}^{\sqrt{14}}$ |

## Appendix 5

## Validation with non-brain objects

We chose to apply our coarse-graining procedure to a range of objects to validate our algorithm, and we included both negative and positive controls.

As a first positive control, we used a simple box with finite thickness (i.e. we simulated a inner 'white matter' surface and an outer 'grey matter' surface, both being a cube, one positioned inside the other). A schematic is shown in *Appendix 5—figure 1*. We know based on theoretical consideration that this box must be aligned exactly with the $K = -\frac{1}{9}S$ line, as its $A_t$ must be the same as its $A_e$ at every stage of coarse-graining, which also implies a 'fractal dimension' of 1. This is exactly the case in our plot of $K$ against $S$, as all the grey data points from the coarse-graining algorithm align on $K = -\frac{1}{9}S$ (thick black line).

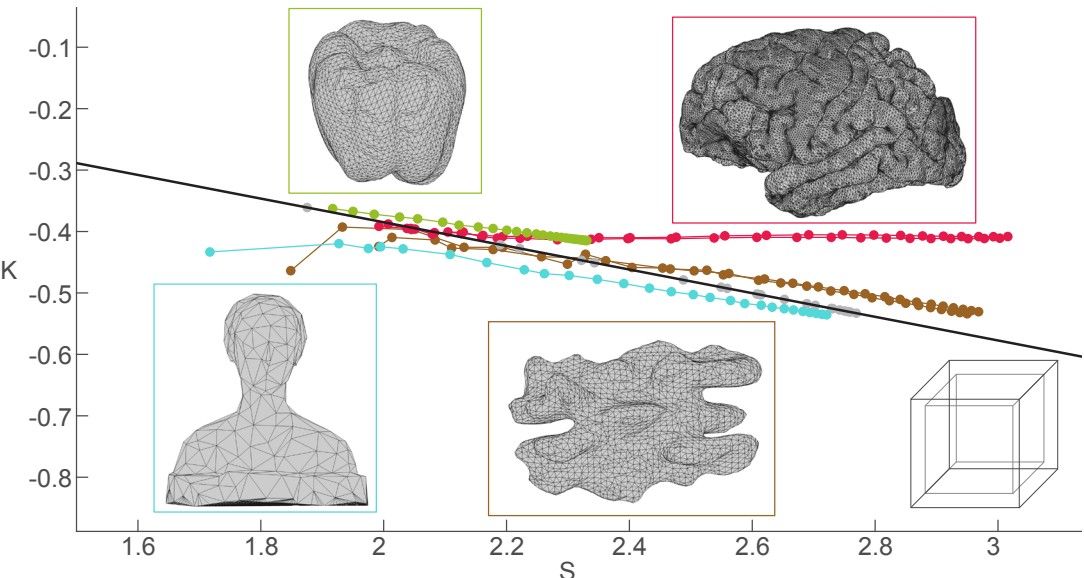

**Appendix 5—figure 1.** Scaling behaviour of various objects in $K \times S$ space. Thick black line indicates $K = -\frac{1}{9}S$ and a gyrification index. $g = \frac{A_t}{A_e} = 1$ Points above and below the line have, respectively, $g > 1$ and $g < 1$. Colour-coded boxes show the surface of the objects we analysed. For simplicity we show the outer 'pial' surface of each object. In the case of the box, we indicated both outer and inner surfaces. Green data points correspond to the bell pepper, red to the human brain, cyan to the 'Laurana' bust, brown to the walnut, and grey to the box of finite thickness.

As negative controls, we included three non-brain objects, a bell pepper, two walnut halves, and a coarse outline of a bust/figurine known as Laurana. All three of these objects show a self-similar, or fractal, regime (corresponding to partially straight trajectories in $K \times S$) in our coarse-graining procedure, but their trajectories in $K \times S$ space are not flat (*Appendix 5—figure 1*), meaning their fractal dimensions are distinct from 2.5. Their trajectories also do not overlap with each other, indicating that they are fundamentally different shapes.

The human brain is included here as a reference and shows a clear flat trajectory (*Appendix 5—figure 1*) as presented in the main text. Also worth noting, after the human trajectory intersects the $K = -\frac{1}{9}S$ line, it veers off to follow the line closely, indicating that the $A_e$ and $A_t$ remain the same with further steps of coarse-graining. Effectively, the human brain transitions to be lissencephalic convex structures once their fractal regime ends. For simplicity, we excluded these lissencephalic data points of extreme coarse-graining from the results in the main text.

## Appendix 6

## Validation with randomising grid for coarse-graining

To assess robustness of the coarse-graining algorithm, we ran 30 different realisations of the algorithm, but with a small random shift of the grid position relative to the surface meshes. The shift is chosen to be within the radius of $\lambda$. This allows subtle changes in the voxelisation at the boundary of grey matter and white matter. Interestingly, over five different human individuals, we observed subtle changes in the values $S$ over the 30 realisations, and to a lesser degree $K$ (*Appendix 6—figure 1*). Additionally, the variation between individuals, especially in $K$ at smaller spatial scales, is far greater than the variation introduced by the jittered realisations. These results suggest that the coarse-graining algorithm is extremely robust towards subtle changes at the grey or white matter boundary, especially far away from the lissencepahlic limit.

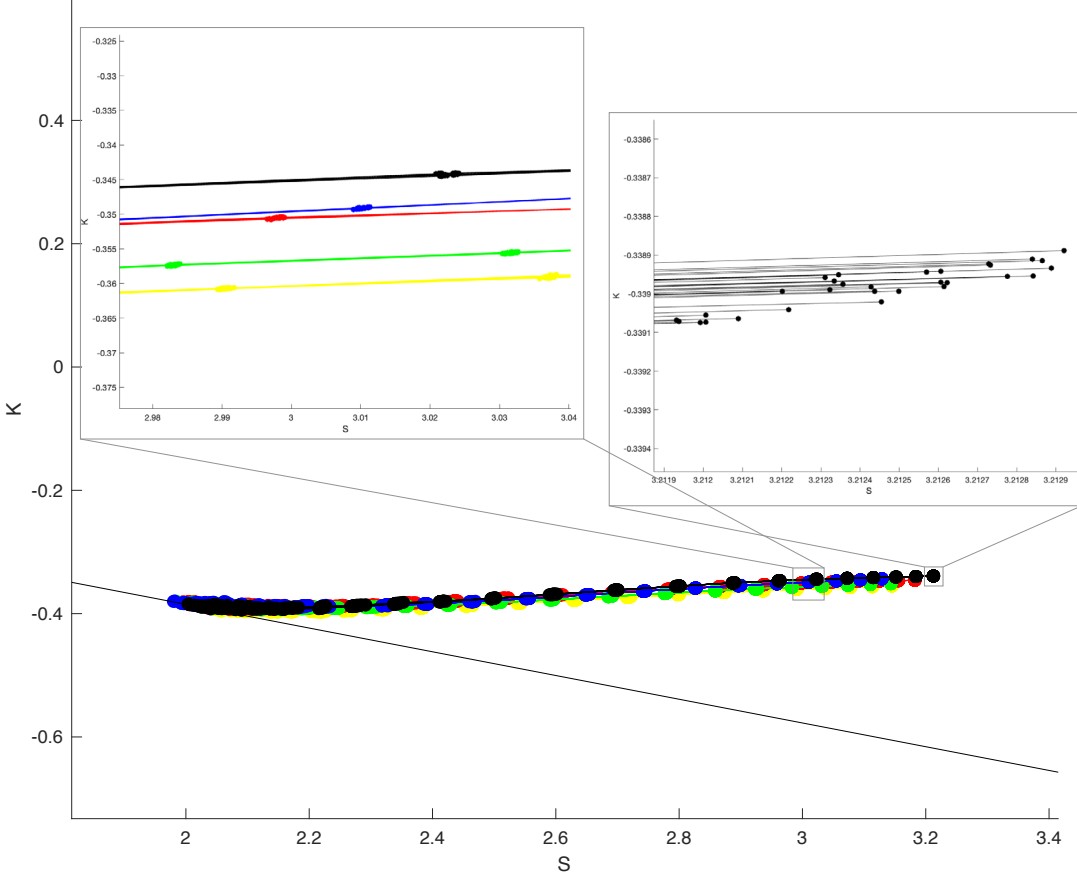

**Appendix 6—figure 1.** Randomising grid for coarse-graining yields very consistent results within individuals in $K \times S$ space. Thick black line indicates $K = -\frac{1}{9}S$ and a gyrification index $g = \frac{A_t}{A_e} = 1$. Each colour in the plot indicates a different human individual (Human Connectome Project, HCP 103414 - yellow, 135225 - red, 138534 - green, 144832 - blue, 148840 - black, respectively). 30 jittered grid outputs are shown for each individual, lines connect datapoints of the same jittered version. Zoomed in panels show that the jittered outputs have far less variation within than between individuals.

## Appendix 7

### Ageing process

#### 7.1 All morphometric variables

In the main text, we used the ageing process as an example for scale-dependent biological process. For completeness, here, we also show all the morphometric variables across scales in the 20 y.o. and 80 y.o. cohort in **Appendix 7—figure 1**, and the corresponding effect sizes in **Appendix 7— figure 2**. As expected, $A_e$ shows very little difference between the two cohorts. $I$, $S$, and $T$ show some effects at scales smaller than 4 mm, but the effect decreases monotonously for higher scales. $K$ demonstrates a more complex scale-dependent effect: larger $K$ in younger subjects at small scales of 0.25 mm (in agreement with previous native scale analyses **Wang et al., 2016**), smaller $K$ in younger subjects at scales of approx. 2 mm, and a return to larger $K$ in younger subjects at large scales of more than approx. 5 mm. Note, however, that despite these large effect sizes, the actual change of values of $K$ is within the range of variation expected across species (seen in the main text). The range of variation of $K$ is approx 0.04 here, and at least an order of magnitude smaller than the range of variation in $S$ (approx 1.5).

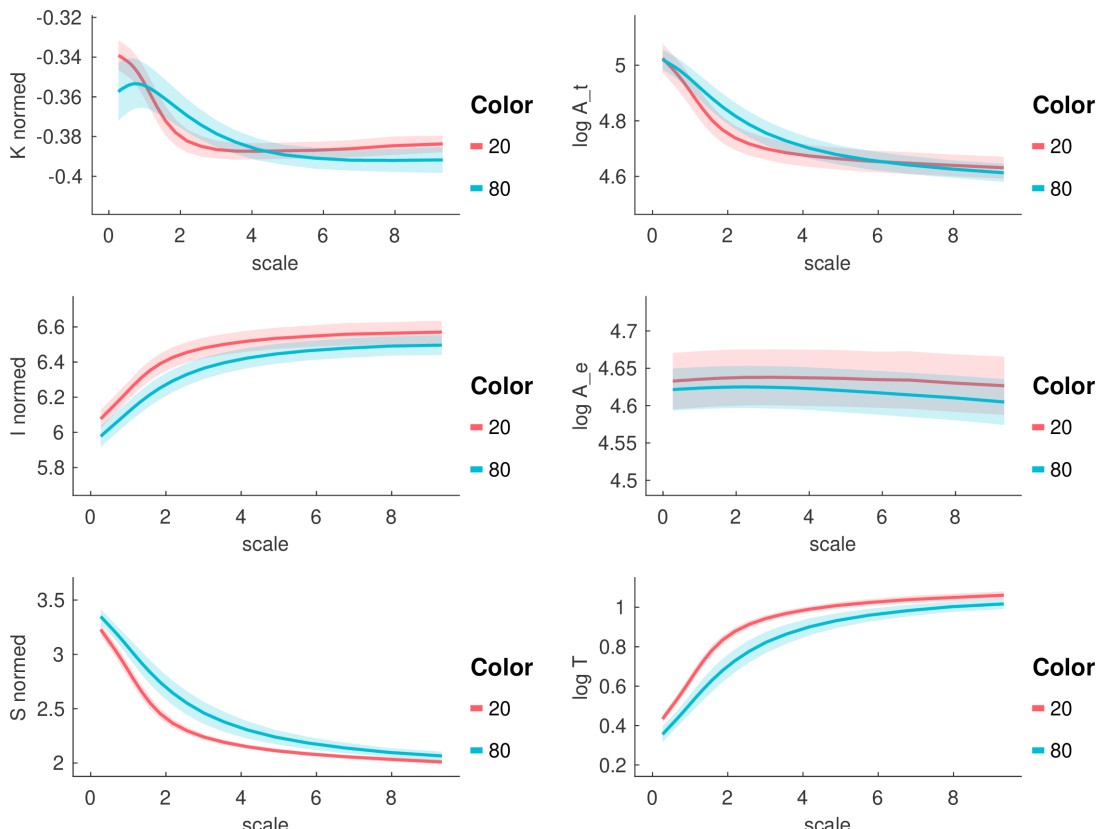

**Appendix 7—figure 1.** All morphological variables across scales in the 20 y.o. and 80 y.o. cohort. Each panel shows a morphological metric, and data is shown for a group of 20-year-olds (red, n=27) and a group of 80-year-olds (blue, n=86). Mean and standard deviation are shown as the solid line and the shaded area, respectively.

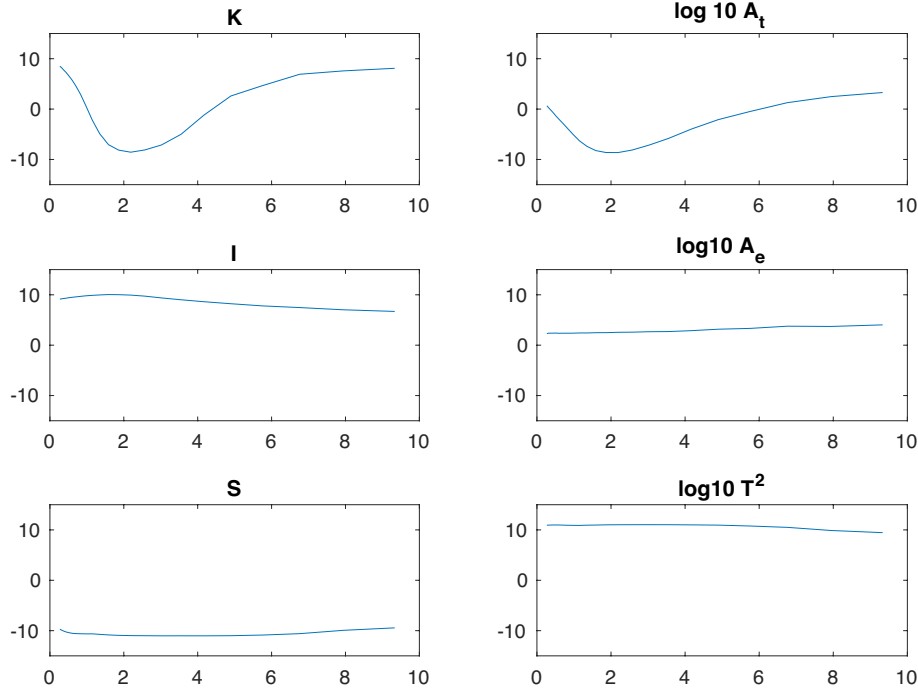

**Appendix 7—figure 2.** Effect size between the 20 y.o. and 80 y.o. cohort in all morphological variables. Each panel shows a morphological metric, and data is shown for a group of 20-year-olds (red, n=27) and a group of 80-year-olds (blue, n=86). Effect size is measured as the rank sum z statistic between the two groups.

## 7.2 Confirmation in independent dataset

To confirm that our observed ageing effect was not driven by sample or data-specific properties, we also analysed an independent dataset (NKI) with the same methods. Here, we show the equivalent figures to *Appendix 7—figure 1* and *Appendix 7—figure 2* for the NKI data in *Appendix 7—figure 3* and *Appendix 7—figure 4*. The same qualitative patterns can be seen in all plots, including the scale-specific effects in $K$ and $A_e$ at around 2 mm.

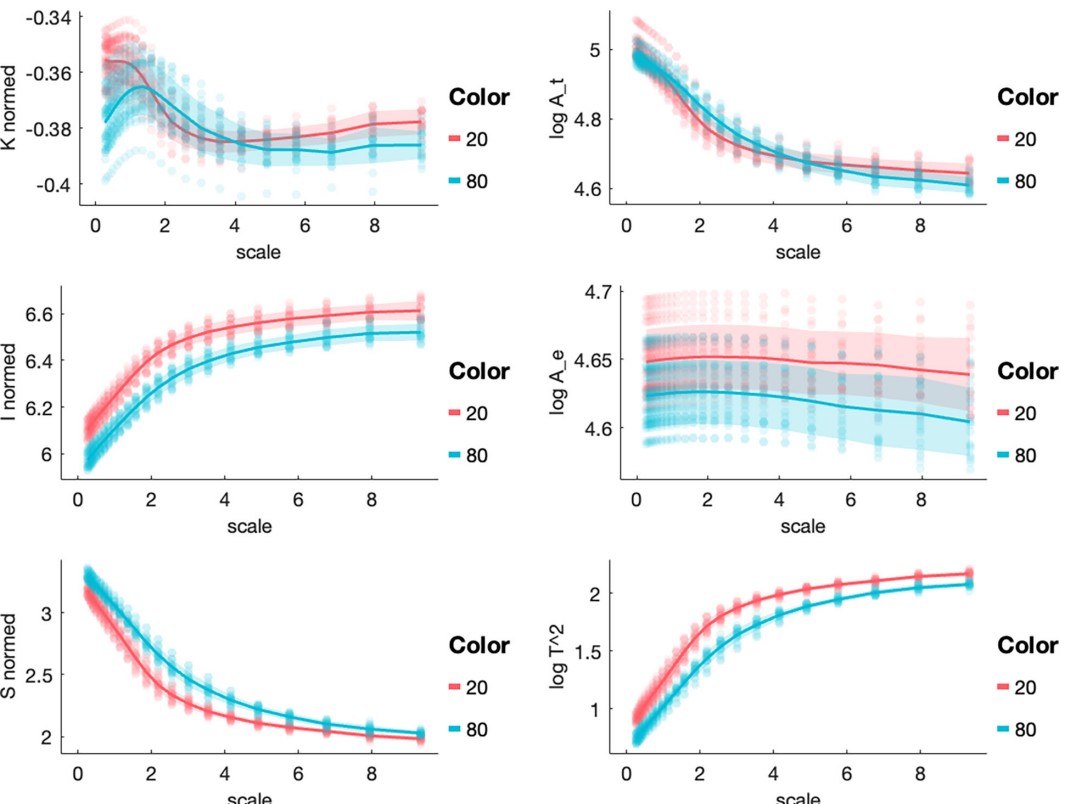

**Appendix 7—figure 3.** All morphological variables across scales in the 20 y.o. and 80 y.o. cohort in a separate (NKI) dataset. Each panel shows a morphological metric, and data from an independent dataset (NKI) is shown for a group of 20-year-olds (red, n=10) and a group of 80-year-olds (blue, n=10).

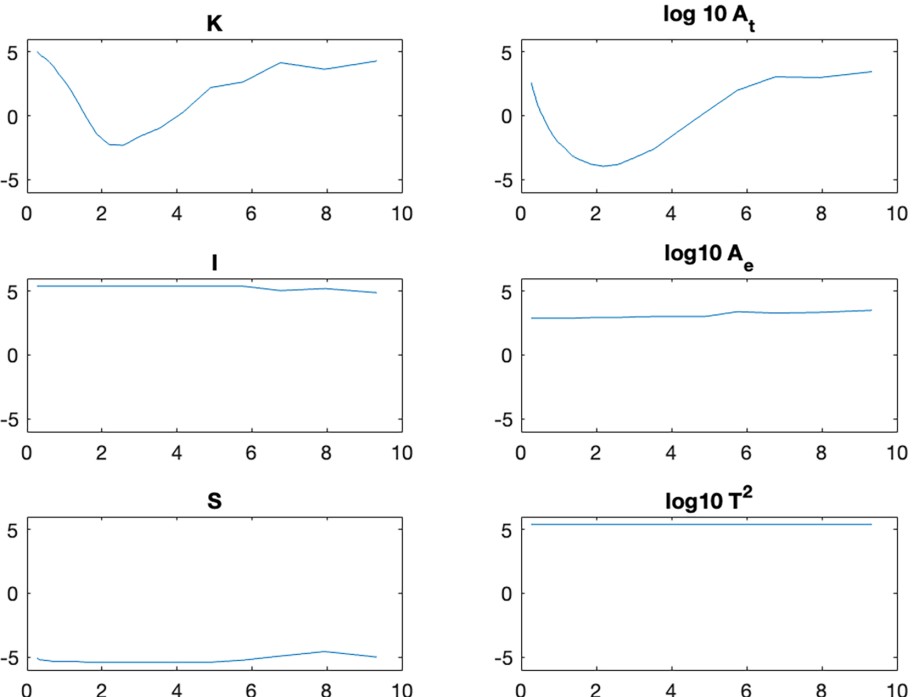

**Appendix 7—figure 4.** Effect size between the 20 y.o. and 80 y.o. cohort in all morphological variables in a separate (NKI) dataset. Effect size is measured as the rank sum z statistic between the two groups. Mean and standard deviation are shown as the solid line and the shaded area, respectively.

