## [Editor Report · eLife assessment]

This study presents **valuable** framework and findings to our understanding of the brain cortex as a fractal object. Based on detailed methodology, the evidence provided on the stability of its shape property within 11 primate species is **convincing**, as well as the scale-specific effects of ageing on the human brain. This study will be of interest to neuroscientists interested in brain morphology, and to physicists and mathematicians interested in modeling the shapes of complex objects.

---

## [Referee Report · Reviewer #2 (Public review)]

In this manuscript, the authors analyze the shapes of cerebral cortices from several primate species, including subgroups of young and old humans, to characterize commonalities in patterns of gyrification, cortical thickness, and cortical surface area. The authors state that the observed scaling law shares properties with fractals, where shape properties are similar across several spatial scales. One way the authors assess this is to perform a "cortical melting" operation that they have devised on surface models obtained from several primate species. The authors also explore differences in shape properties between brains of young (~20 year old) and old (~80) humans. A challenge the authors acknowledge struggling with in reviewing the manuscript is merging "complex mathematical concepts and a perplexing biological phenomenon." This reviewer remains a bit skeptical about whether the complexity of the mathematical concepts being drawn from are justified by the advances made in our ability to infer new things about the shape of the cerebral cortex.

(1) The series of operations to coarse-grain the cortex illustrated in Figure 1 produces image segmentations that do not resemble real brains. The process to assign voxels in downsampled images to cortex and white matter is biased towards the former, as only 4 corners of a given voxel are needed to intersect the original pial surface, but all 8 corners are needed to be assigned a white matter voxel. The reason for introducing this bias (and to the extent that it is present in the authors' implementation) is not provided. The authors provide an intuitive explanation of why thickness relates to folding characteristics, but ultimately an issue for this reviewer is, e.g., for the right-most panel in Figure 2b, the cortex consists of several 4.9-sided voxels and thus a >2 cm thick cortex. A structure with these morphological properties is not consistent with the anatomical organization of typical mammalian neocortex.

(2) For the comparison between 20-year-old and 80-year-old brains, a well-documented difference is that the older age group possesses more cerebral spinal fluid due to tissue atrophy, and the distances between the walls of gyri becomes greater. This difference is born out in the left column of Figure 4b. It seems this additional spacing between gyri in 80 year olds requires more extensive down-sampling (larger scale values in Figure 4a) to achieve a similar shape parameter K as for the 20 year olds. The authors assert that K provides a more sensitive measure (associated with a large effect size) than currently used ones for distinguishing brains of young vs. old people. A more explicit, or elaborate, interpretation of the numbers produced in this manuscript, in terms of brain shape, might make this analysis more appealing to researchers in the aging field.

(3) In the Discussion, it is stated that self-similarity, operating on all length scales, should be used as a test for existing and future models of gyrification mechanisms. Given the lack of association between the abstract mathematical parameters described in this study and explicit properties of brain tissue and its constituents, it is difficult to envision how the coarse-graining operation can be used to guide development of "models of cortical gyrification."

(4) There are several who advocate for analyzing cortical mid-thickness surfaces, as the pial surface over-represents gyral tips compared to the bottoms of sulci in the surface area. The authors indicate that analyses of mid-thickness representations will be taken on in future work, but this seems to be a relevant control for accepting the conclusions of this manuscript.

---

## [Referee Report · Reviewer #3 (Public review)]

Summary: Through a rigorous methodology, the authors demonstrated that within 11 different primates, the shape of the brain followed a universal scaling law with fractal properties. They enhanced the universality of this result by showing the concordance of their results with a previous study investigating 70 mammalian brains, and the discordance of their results with other folded objects that are not brains. They incidentally illustrated potential applications of this fractal property of the brain by observing a scale-dependant effect of aging on the human brain.

Strengths:

- New hierarchical way of expressing cortical shapes at different scales derived from previous report through implementation of a coarse-graining procedure

- Investigation of 11 primate brains and contextualisation with other mammals based on prior literature

- Proposition of tool to analyse cortical morphology requiring no fine tuning and computationally achievable

- Positioning of results in comparison to previous works reinforcing the validity of the observation.

- Illustration of scale-dependance of effects of brain aging in the human.

Weaknesses:

- The notion of cortical shape, while being central to the article, is not really defined, leaving some interpretation to the reader

- The organization of the manuscript is unconventional, leading to mixed contents in different sections (sections mixing introduction and method, methods and results, results and discussion...). As a result, the reader discovers the content of the article along the way, it is not obvious at what stages the methods are introduced, and the results are sometimes presented and argued in the same section, hindering objectivity.

To improve the document, I would suggest a modification and restructuring of the article such that: (1) by the end of the introduction the reader understands clearly what question is addressed and the value it holds for the community, (2) by the end of the methods the reader understands clearly all the tools that will be used to answer that question (not just the new method), (3) by the end of the results the reader holds the objective results obtained by applying these tools on the available data (without subjective interpretations and justifications), and (4) by the end of the discussion the reader understands the interpretation and contextualisation of the study, and clearly grasps the potential of the method depicted for the better understanding of brain folding mechanisms and properties.

---

## [Author Response]

The following is the authors’ response to the previous reviews.

**eLife assessment:**
This study presents valuable framework and findings to our understanding of the brain as a fractal object by observing the stability of its shape property within 11 primate species and by highlighting an application to the effects of aging on the human brain. The evidence provided is solid but the link between brain shape and the underlying anatomy remains unclear. This study will be of interest to neuroscientists interested in brain morphology, whether from an evolutionary, fundamental or pathological point of view, and to physicists and mathematicians interested in modeling the shapes of complex objects.

We now clarified the outstanding questions regarding if our model outputs can be related to actual primate brain anatomy, which we believe was mainly based on comments regarding the validity of our output of apparently thicker cortices than nature can produce.

We address this point in more detail in the point-by-point response below, but want to address this misunderstanding directly here: Our algorithm **does not** produce thicker cortices with increasing coarse-graining scales; in fact, the cortical thickness never exceeds the actual cortical thickness in our outputs, but rather thins with each coarse-graining scale. In other words, we believe that our outputs are fully in line with neuroanatomy across species.

**Reviewer #2 (Public Review):**
In this manuscript, the authors analyze the shapes of cerebral cortices from several primate species, including subgroups of young and old humans, to characterize commonalities in patterns of gyrification, cortical thickness, and cortical surface area. The authors state that the observed scaling law shares properties with fractals, where shape properties are similar across several spatial scales. One way the authors assess this is to perform a "cortical melting" operation that they have devised on surface models obtained from several primate species. The authors also explore differences in shape properties between brains of young (~20 year old) and old (~80) humans. A challenge the authors acknowledge struggling with in reviewing the manuscript is merging "complex mathematical concepts and a perplexing biological phenomenon." This reviewer remains a bit skeptical about whether the complexity of the mathematical concepts being drawn from are justified by the advances made in our ability to infer new things about the shape of the cerebral cortex.

To allow scientists from all backgrounds to adopt these complex ideas, we have made our code to “melt” the brains and for further downstream analysis publicly available. We have now also provided a graphical user interface, to allow users without substantial coding experience to run the analysis. We also believe that the algorithmic concepts are easy to understand due to the similarity to the coarse-graining procedures found in long-standing and well-accepted box-counting algorithms.

Beyond the theoretical insight of the fractal nature of cortices and providing an explicit and crucial link between vastly different brains that are gyrified and those that are not, we believe that the advance gained by our methods for future applications is clearly demonstrated in our proof-of-principle with a four-fold increase in effect size. For reference, an effect size of 8 would translate to an almost perfect separation of groups, i.e. an ideal biomarker with near 100% sensitivity and specificity.

(1) The series of operations to coarse-grain the cortex illustrated in Figure 1 produces image segmentations that do not resemble real brains.

As re-iterated in our Methods and Discussion: “Note, of course, that the coarse-grained brain surfaces are an output of our algorithm alone and are not to be directly/naively likened to actual brain surfaces, e.g. in terms of the location or shape of the folds. Our comparisons here between coarse-grained brains and actual brains is purely on the level of morphometrics across the whole cortex.”

Fig. 1 therefore serves as an explanation to the reader on the algorithmic outputs, but each melted brain is not supposed to be directly/visually compared to actual brains. Similar to algorithms measuring the fractal dimension, or the exposed surface area of a given brain, the intermediate outputs of these algorithms are not supposed to represent any biologically observed brain structures, but rather serve as an abstraction to obtain meaningful morphometrics.

We additionally added a note to the caption of Fig. 1 to clarify this point:

“Note that the actual size of the brains for analysis are rescaled (see Methods and Fig. 3); we display all brains scaled at an equal size here for the ease of visualisation of the method.”

Finally, we also edited the entire paper for terminology to clearly distinguish the terms of (1) the cortex as a 3D object, (2) coarse-grained and voxelised versions thereof, and (3) summary morphological measures derived from the former. When we invite comparisons in our paper between real brains and coarse-grained brains, this is always at the level of summary morphological measures, not at the level of the 3D objects/voxelisations themselves.

The process to assign voxels in downsampled images to cortex and white matter is biased towards the former, as only 4 corners of a given voxel are needed to intersect the original pial surface, but all 8 corners are needed to be assigned a white matter voxel. The reason for introducing this bias (and to the extent that it is present in the authors' implementation) is not provided.

This detail was in the Supplementary, and we have now added additional clarification on this specific point to our Supplementary:

“In detail, we assign all voxels in the grid with at least four corners inside the original pial surface to the pial voxelization. This process allows the exposed surface to remain approximately constant with increasing voxel sizes. A constant exposed surface is desirable, as we only want to gradually ‘melt’ and fuse the gyri, but not grow the bounding/exposed surface as well. We want the extrinsic area to remain approximately constant as we decrease the intrinsic area via coarse-graining; it is like generating iterates of a Koch curve in reverse, from more to less detailed, by increasing the length of smallest line segment.

We then assign voxels with all eight corners inside the original white matter surface to the white matter voxelization. This is to ensure integrity of the white matter, as otherwise white matter voxels in gyri may become detached from the core white matter, and thus artificially increase white matter surface area. Indeed, the main results of the paper are not very sensitive to this decision using all eight corners, vs. e.g. only four corners, as we do not directly use white matter surface area for the scaling law measurements. However, we still maintained this choice in case future work wants to make use of the white matter voxelisations or derivative measures.”

Note on the point of white matter integrity that if both grey and white matter voxelisations require all 8 corner to be inside the respective mesh, there will be voxels not assigned to either at the grey/white matter interface, causing potential downstream issues.

We further acknowledge:

“Of course, our proposed procedure is not the only conceivable way to erase shape details below a given scale; and we are actively working on related algorithms that are also computationally cheaper. Nevertheless, the current version requires no fine-tuning, is computationally feasible and conceptually simple, thus making it a natural choice for introducing the methodology and approach.”

The authors provide an intuitive explanation of why thickness relates to folding characteristics, but ultimately an issue for this reviewer is, e.g., for the right-most panel in Figure 2b, the cortex consists of several 4.9-sided voxels and thus a >2 cm thick cortex. A structure with these morphological properties is not consistent with the anatomical organization of typical mammalian neocortex.

We assume the reviewer refers to Fig. 1B with the panel on scale=4.9mm. We would like to point out that Fig. 1 serves as an explanation of the voxelisation method. For the actual analysis and Results, we are using re-scaled brains (see Fig. 2 with the ever decreasing brain sizes). The rescaling procedure is now expanded as below:

“Morphological properties, such as cortical thicknesses measured in our ‘melted’ brains are to be understood as a thickness relative to the size of the brain. Therefore, to analyse the scaling behaviour of the different coarse-grained realisations of the same brain, we apply an isometric rescaling process that leaves all dimensionless shape properties unaffected (more details in Suppl. S3.1). Conceptually, this process fixes the voxel size, and instead resizes the surfaces relative to the voxel size, which ensures that we can compare the coarse-grained realisations to the original cortices, and test if the former, like the latter, also scale according to Eqn. (1). Resizing, or more precisely, shrinking the cortical surface is mathematically equivalent to increasing the box size in our coarse-graining method. Both achieved an erasure of folding details below a certain threshold. After rescaling, as an example, the cortical thickness also shrinks with increasing levels of coarse-graining, and never exceeds the thickness measured at native scale.”

We additionally added a note to the caption of Fig. 1 to clarify this point:

“Note that the actual size of the brains for analysis are rescaled (see Methods and Fig. 3); we display all brains scaled at an equal size here for the ease of visualisation of the method.”

Finally, we also edited the entire paper for terminology to clearly distinguish the terms of (1) the cortex as a 3D object, (2) coarse-grained versions thereof, and (3) summary morphological measures derived from the former. When we invite comparisons in our paper between real brains and coarse-grained brains, this is always at the level of summary morphological measures, not at the level of the 3D objects themselves and their detailed anatomical features.

(2) For the comparison between 20-year-old and 80-year-old brains, a well-documented difference is that the older age group possesses more cerebral spinal fluid due to tissue atrophy, and the distances between the walls of gyri becomes greater. This difference is born out in the left column of Figure 4b. It seems this additional spacing between gyri in 80 year olds requires more extensive down-sampling (larger scale values in Figure 4a) to achieve a similar shape parameter K as for the 20 year olds. The authors assert that K provides a more sensitive measure (associated with a large effect size) than currently used ones for distinguishing brains of young vs. old people. A more explicit, or elaborate, interpretation of the numbers produced in this manuscript, in terms of brain shape, might make this analysis more appealing to researchers in the aging field.

We have removed the main results relating to K and aging from our last revision already to avoid confusion. This is now only in the supplementary analysis, and our claim of K being a more sensitive measure for age and ageing – whilst still true – will be presented in more detail in a series of upcoming papers.

(3) In the Discussion, it is stated that self-similarity, operating on all length scales, should be used as a test for existing and future models of gyrification mechanisms. Given the lack of association between the abstract mathematical parameters described in this study and explicit properties of brain tissue and its constituents, it is difficult to envision how the coarse-graining operation can be used to guide development of "models of cortical gyrification."

We have clarified in more detail what we meant originally in Discussion:

“Finally, this dual universality is also a more stringent test for existing and future models of cortical gyrification mechanisms at relevant scales, and one that moreover is applicable to individual cortices. For example, any models that explicitly simulate a cortical surface as an output could be directly coarse-grained with our method and the morphological trajectories can be compared with those of actual human and primate cortices. The simulated cortices would only be ‘valid’ in terms of the dual universality, if it also produces the same morphological trajectories.”

However, we agree with the reviewer that our paper could be misread as demanding direct comparisons of each coarse-grained brain with an actual brain, and we have now added the following text to clarify that this is not our intention for the proposed method or outputs.

“Note, we do not suggest to directly compare coarse-grained brain surfaces with actual biological brain surfaces. As we noted earlier, the coarse-grained brain surfaces are an output of our algorithm alone and not to be directly/naively likened to actual brain surfaces, e.g. in terms of the location or shape of the folds. Our comparisons here between coarse-grained brains and actual brains is purely on the level of morphometrics across the whole cortex.”

Indeed, the dual universality imposes restrictive constraints on the possible shapes of real cortices, but do not fully specify them. Presumably, the location of individual folds in different individuals and species will depend on their respective evolutionary histories, so there is no reason to expect a match in fold location between the ‘melted’ cortices of more gyrified species, on one hand, and the cortex of a less-gyrified one, on the other, even if their global morphological parameters and global mechanism of folding coincide.

(4) There are several who advocate for analyzing cortical mid-thickness surfaces, as the pial surface over-represents gyral tips compared to the bottoms of sulci in the surface area. The authors indicate that analyses of mid-thickness representations will be taken on in future work, but this seems to be a relevant control for accepting the conclusions of this manuscript.

In the context of some applications and methods, we agree that the mid-surface is a meaningful surface to analyse. However, in our work, the mid-surface is not. The fractal estimation rests on the assumption that the exposed area hugs the object of interest (hence convex hull of the pial surface), as the relationship between the extrinsic and intrinsic areas across scales determine the fractal relationship (Eq. 2). If we used the mid-surface instead of the pial surface for all estimation, this would not represent the actual object of interest, and it is separated from the convex hull. Estimating a new convex hull based on the mid surface would be the equivalent of asking for the fractal dimension of the mid-surface, not of the cortical ribbon. In other words, it would be a different question, bound to yield a different answer.

Hence, we indicated in our original response that we only have a provisional answer, but more work beyond the scope of this paper is required to answer this question, as it is a separate question. The mid-surface, as a morphological structure in its own right, will have its own scaling properties, and our provisional understanding is that these also yield a scaling law parallel to those of the cortical ribbon with the same or a similar fractal dimension. But more systematic work is required to investigate this question at native scale and across scales.

**Reviewer #3 (Public Review):**
Summary: Through a rigorous methodology, the authors demonstrated that within 11 different primates, the shape of the brain followed a universal scaling law with fractal properties. They enhanced the universality of this result by showing the concordance of their results with a previous study investigating 70 mammalian brains, and the discordance of their results with other folded objects that are not brains. They incidentally illustrated potential applications of this fractal property of the brain by observing a scale-dependant effect of aging on the human brain.Strengths:- New hierarchical way of expressing cortical shapes at different scales derived from previous report through implementation of a coarse-graining procedure- Investigation of 11 primate brains and contextualisation with other mammals based on prior literature- Proposition of tool to analyse cortical morphology requiring no fine tuning and computationally achievable- Positioning of results in comparison to previous works reinforcing the validity of the observation.- Illustration of scale-dependance of effects of brain aging in the human.Weaknesses:- The notion of cortical shape, while being central to the article, is not really defined, leaving some interpretation to the reader- The organization of the manuscript is unconventional, leading to mixed contents in different sections (sections mixing introduction and method, methods and results, results and discussion...). As a result, the reader discovers the content of the article along the way, it is not obvious at what stages the methods are introduced, and the results are sometimes presented and argued in the same section, hindering objectivity.To improve the document, I would suggest a modification and restructuring of the article such that: (1) by the end of the introduction the reader understands clearly what question is addressed and the value it holds for the community, (2) by the end of the methods the reader understands clearly all the tools that will be used to answer that question (not just the new method), (3) by the end of the results the reader holds the objective results obtained by applying these tools on the available data (without subjective interpretations and justifications), and (4) by the end of the discussion the reader understands the interpretation and contextualisation of the study, and clearly grasps the potential of the method depicted for the better understanding of brain folding mechanisms and properties.

We thank this reviewer again for their attention to detail and constructive comments. We have followed the detailed suggestions provided by us in the Recommendations For The Authors, and summarise the main changes here:

- We have restructured all sections to be more clearly following Introduction, Methods, Results, and Discussion; by using subsections, we believe the structure is now more accessible to readers.

- We have now clarified the concept of “cortical shape”, as we use it in our paper in several places, by distinguishing clearly the object of study, and the morphological properties measured from it.

**Recommendations for the authors:**
**Reviewer #2 (Recommendations For The Authors):** None
**Reviewer #3 (Recommendations For The Authors):**
I once again compliment the authors for their elegant work. I am happy with the way they covered my first feedback. My second review takes into account some comments made by other reviewers with which I agree.

We thank this reviewer again for their attention to detail and constructive comments.

Recommendations for clarifications:General comments: The purpose of the article could be made clearer in the introduction. When I differentiate results from discussion, I think of results as objective measures or observations, while discussion will relate to the interpretation of these results (including comparison with previous literature, in most cases).

We have restructured all sections to be more clearly following Introduction, Methods, Results, and Discussion; by using subsection, we believe the structure is now more accessible to readers.

- l.39: define or discuss "cortical shape"

We have gone through the entire paper and corrected for any ambiguities. We specifically distinguish between the cortex as a structure overall, shape measures derived from this structure, and coarse-grained versions of the structure.

- l.48-74: this would match either an introduction or a discussion rather than a methods section.

Done

- l.98-106: this would match a discussion rather than a methods section.

Done

- l.111: here could be a good spot to discuss the 4 vs 8 corners for inclusion of pial vs white matter voxelization

We have discussed this in the more detailed Supplementary section now, as after restructuring, this appears to be the more suitable place.

- l.140-180: it feels that this section mixes methods, results and discussion of the results

We agree and we have resolved this by removing sentences and re-arranging sections.

- l.183-217: mix of results and discussion

We agree and we have resolved this by removing sentences and re-arranging sections.

Small cosmetic suggestions:- l.44: conservation of 'some' quantities: vague

Changed to conservation of morphological relationships across evolution

- l.66: order of citations ([24, 22,23])

Will be fixed at proof stage depending on format of references.

- l.77: delete space between citation and period

Done

- l.77: I would delete 'say'

Done

- l.86: 'but to also analyse' -> 'to analyse'

Done

- l.105: remove 'we are encouraged that'

Done

- l.111: 'also see' -> 'see also'

Done

- l.164: 'remarkable': subjective

Done

- l.189: define approx. abbreviation

Done

- l.190: 'approx' -> 'approx.'

Revised

- l.195: 'dramatic': subjective

removed

-l. 246: 'much' -> vague

explained